# B-MoCA: Benchmarking Mobile Device Control Agents across Diverse Configurations

## Abstract

Mobile device control agents can largely enhance user interactions and productivity by automating daily tasks. However, despite growing interest in developing practical agents, the absence of a commonly adopted benchmark in this area makes it challenging to quantify scientific progress. In this work, we introduce B-MoCA: a novel benchmark with interactive environments for evaluating and developing mobile device control agents. To create a realistic benchmark, we develop B-MoCA based on the Android operating system and define 131 common daily tasks. Importantly, we incorporate a randomization feature that changes the configurations of mobile devices, including user interface layouts and language settings, to assess generalization performance. We benchmark diverse agents, including agents employing large language models (LLMs) or multi-modal LLMs as well as agents trained with imitation learning using human expert demonstrations. While these agents demonstrate proficiency in executing straightforward tasks, their poor performance on complex tasks highlights significant opportunities for future research to improve effectiveness.

## 1 Introduction

Autonomous agents controlling mobile devices have great potential benefits. For example, these agents can improve the accessibility of user interactions, especially for users with physical disabilities or those facing challenges in operating devices. Additionally, they can boost productivity by automating daily tasks. Such advantages have led to increased interest in developing practical agents for *mobile* device control. Various approaches have been introduced, including agents based on large language models (LLMs) (Wen et al., 2023; Yan et al., 2023) and agents trained with human demonstrations (Sun et al., 2022; Li et al., 2023; Rawles et al., 2023). These innovations aim to create assistive agents capable of understanding device screen layouts and manipulating user interfaces (UI) to execute human commands.

Despite recent progress in developing mobile device control agents based on real systems, such as Android emulators (Toyama et al., 2021; Shvo et al., 2021; Zhang et al., 2023), prior works often overlook several important properties. One primary aspect is testing the generalization ability across diverse device configurations, which is crucial in deploying agents in real devices. Moreover, practical tasks essential for life (such as setting an alarm or making emergency calls) are often neglected because of the challenges in defining a wide range of such tasks with robust success criteria in various device settings. The lack of a unified benchmark encompassing these important properties has impeded scientific progress in this field.

In this work, we introduce B-MoCA: a **B**enchmark designed for evaluating **Mo**bile device **C**ontrol **A**gents across diverse configurations (see Figure 1). For real-system interactive evaluation, B-MoCA is developed based on the Android operating system. A key feature of B-MoCA is supporting numerous customization, designed to mirror diverse device configurations in real-use cases, including variations in icon placements, sizes, wallpapers, languages, and device types. Utilizing this feature, one can easily create diverse environments with various configurations to evaluate the agents' generalization ability. Additionally, we define 131 practical tasks grounded in realistic scenarios, such as opening specific applications, initializing searches over the web, and adjusting device settings. To ensure reliable evaluation across diverse configurations, B-MoCA provides rule-based success detectors that automatically signal task completion during the agents' interactions over the environments.

Figure 1: Illustration of B-MoCA. We present a realistic benchmark for assessing the performances of mobile device control agents in executing everyday tasks. A key feature of B-MoCA is supporting randomization that changes various device attributes to analyze generalization ability. We benchmark agents leveraging LLMs or MLLMs as well as custom agents trained from scratch.

We benchmark various methods for building mobile device control agents in B-MoCA. The baselines include agents employing foundation models, such as large language models (LLMs) or multi-modal LLMs (MLLMs), which benefit from extensive knowledge obtained through pre-training. We consider both closed-source models, such as GPT-4o (OpenAI, 2024) and Gemini-1.5-pro (Gemini et al., 2023), and open-source models, such as Llama-3 (Meta, 2024). Additionally, we consider building agents by training policies using behavior cloning (BC) (Pomerleau, 1988; Schaal, 1996).

In our experiments, we find that the tested agents demonstrate capabilities in solving straightforward tasks. However, agents employing foundation models (like LLMs) show limitations in more challenging scenarios that require multiple interactions. Custom agents trained from scratch successfully mimic expert behaviors but lack the ability to generalize to unseen device configurations. Our extensive experiments reveal the limitations of existing methods, calling for future research.

Our contributions are as follows:

- We propose B-MoCA, a new benchmark designed to measure progress in developing device control agents, including various features such as environment randomization.
- We evaluate several baseline agents for mobile device control, identifying their limitations, such as their poor generalization in UI elements understanding and manipulation.
- We explore different design choices for leveraging foundation models, and analyze the impact of data diversity on the effectiveness of agents trained from scratch.
- We open-source all the source codes and relevant materials for easy reproduction of our environments and experiments.

We hope B-MoCA helps future researchers identify challenges in building assistive agents and easily compare the efficacy of their methods over the prior work.

## 2 B-MoCA

In this section, we introduce B-MoCA, a benchmark designed to develop agents capable of executing common daily tasks on mobile devices with diverse configurations.

### 2.1 DESIGN FACTORS

Designing a meaningful benchmark poses significant challenges, particularly in developing a realistic platform that incorporates practical tasks. Our benchmark is built on Android, a widely used open-

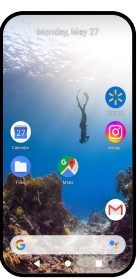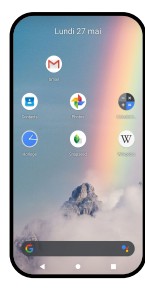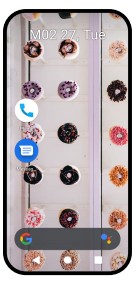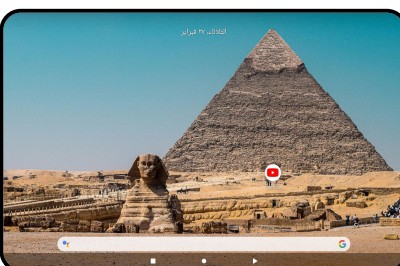

Figure 2: Examples of home screen images from environments in B-MoCA. The randomized features span icon location, font size, wallpaper, language, and device type and challenge the generalization ability of agents.

source operating system, ensuring authentic environments. To reflect the multi-step nature of real interactions, we model the device control task as a sequential decision-making problem (Section 2.2). The benchmark includes 131 tasks involving both default applications like Chrome and Calendar, and third-party applications such as Instagram and Wikipedia, selected for their prevalence and utility in everyday life. Each task is equipped with a success detector to evaluate the agent's performance in accurately completing the task (Section 2.3).

Given the diverse nature of user mobile device setups, such as variations in icon placements, wallpaper choices, languages, and device types, it is important to test the generalization abilities of device control agents across diverse setups. To assess generalization performance, we incorporate a randomization feature in our benchmark. This feature is designed to simulate various real-world scenarios by changing the settings of mobile devices (Section 2.4).

## 2.2 PROBLEM FORMULATION

We formulate the device control task as a sequential decision-making problem, where an agent interacts with an environment (i.e., an Android emulator). Formally, given a task instruction $c$, the agent receives an observation $o_t$ and takes an action $a_t$ based on its policy $\pi(a_t|o_t, c)$ at each timestep $t$. The environment returns a success signal $r_t$ and then transitions to the next observation $o_{t+1}$.

Observations, which capture the UI elements, can be represented as either screen pixels, screen descriptions derived from the Android view hierarchy, or a combination of both. The action space includes a set of screen-touching actions. In B-MoCA, we support both continuous and discrete actions. Continuous actions are defined as dual-gesture actions, similar to Rawles et al. (2023). Each dual-gesture action comprises a pair of $(x, y)$ screen locations. A dual-gesture action is identified as to tap when the two locations are identical within a specified threshold, and as to swipe when the distance between the two locations exceeds this threshold. Also, agents can perform to press navigation buttons (i.e., back, home, and overview) by touching the corresponding locations on the screen. Discrete actions are defined as direct interactions with specific screen locations (such as the center of a UI element's bounding box or predefined locations), swiping in specified directions, or pressing individual buttons. We note that our benchmark supports text-based actions, enabling the utilization of both LLMs and MLLMs (see Section 3.1 for details).

We refer readers for further details on the environment implementation to Appendix A.1.

## 2.3 DAILY TASKS

B-MoCA includes 131 tasks that can be used to assess essential skills for mobile device management. Each task is designed to be grounded in realistic situations to provide functionalities useful in daily routines, such as setting the alarm and enabling airplane mode. These tasks require agents to navigate the device among diverse screens and manipulate various UI elements. Figure 3 shows the statistics of the tasks. For a comprehensive list of tasks, we refer readers to Appendix B.1.

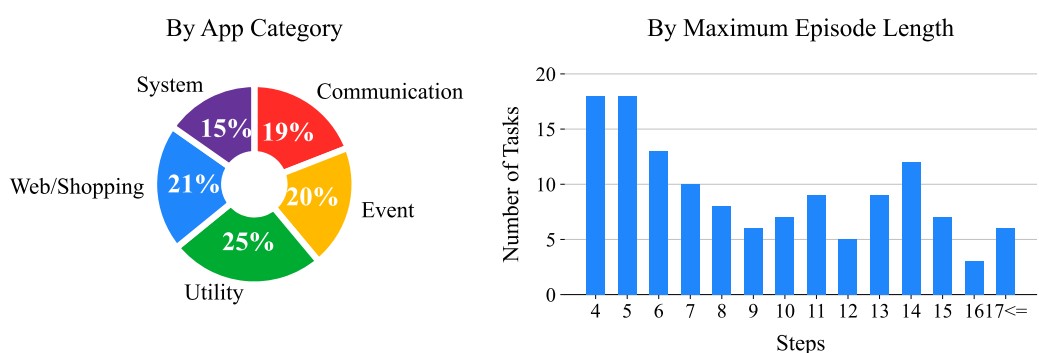

Figure 3: Distribution of tasks by app category and the maximum episode length.

Task completion is determined by a rule-based success detector that relies on three sources: the system information, app data, and attributes of UI elements in the Android view hierarchy. To monitor these sources, we have developed a set of interfaces using tools like Android Debug Bridge (ADB) and Appium. The success detector identifies the successful completion based on pre-defined criteria within these information sources. For example, it automatically detects the matching regular expression in the system logs or checks whether the attributes in specific UI elements (e.g., the 'checked' attribute of the checkbox) are arranged correctly. The success signal has the value of $+1$ when the task is completed, and $0$ otherwise. An episode terminates as a success if the success detector signals completion, or as a failure if the agent exceeds a maximum number of steps without meeting the criteria.

### 2.4 ENVIRONMENT RANDOMIZATION

In mobile device control, developing agents that can generalize across various device setups is crucial. To evaluate their generalization ability, B-MoCA incorporates a randomization feature that changes icon placements and sizes, wallpapers, languages, and device types. Users can select a device type from a device list that includes popular models like Pixel 3, Pixel 4, Pixel 6, and WGXA Tablet. They can also specify the locales to set the language and region, choose wallpapers from a selection of custom images, and activate dark mode for further environmental variation. Moreover, the sizes and locations of application icons can be customized to simulate real-world usage patterns.

Using randomization features, we create 45 unique environments in B-MoCA, with examples shown in Figure 2. To assess the generalization ability, we divide these environments into two sets: 35 for training and 10 for testing. We employ domain randomization (Tobin et al., 2017) to train agents, enabling them to perform tasks robustly across diverse device configurations. We then evaluate the performance on test environments, which include unseen device setups. A detailed list of environment device configurations we prepare is available in Appendix A.2.

## 3 BASELINE AGENTS

In this work, we benchmark various approaches for building mobile device control agents. Section 3.1 describes LLM agents and MLLM agents, where the agents are developed with closed-source LLMs and MLLMs, respectively. In Section 3.2, we introduce custom agents that are equipped with either fine-tuned open-source LLMs or policies trained from scratch.

### 3.1 CLOSED-SOURCE MODELS

Utilizing foundation models like LLMs and MLLMs, which contain extensive knowledge and possess emergent capabilities, has become a major approach in developing mobile device control agents (Wen et al., 2023; Yan et al., 2023). We benchmark two types of agents that employ different foundation models: LLMs (e.g., Llama-3; Meta 2024) and MLLMs (e.g., GPT-4o; OpenAI 2024). LLM agents utilize only the text descriptions of the screen layout to generate text actions, while MLLM agents leverage both text and visual inputs.

| Action option | Description |
|---|---|
| dual-gesture(*) | Operate a dual-gesture action with arguments (*). |
| tap(numeric tag) | Tap UI element labeled with numeric tag. |
| swipe(direction) | Swipe to direction. |
| press("HOME") | Press home button. |
| press("BACK") | Press back button. |
| press("OVERVIEW") | Press overview button. |

Table 1: A set of action options for agents generating text-based actions. The options include both continuous and discrete actions.

**Role**: You are an agent that is trained to perform daily tasks on digital devices, such as smartphones.

**Output format**: Your output should include:
• Observation: Describe what you observe in the input.
• Thought: Provide a rationale on the next action...
• Action: Select an action option in the format of a function...

**Action space**: For the action, you need to select [...]

**Goal**: [...]

(Optional) **Few-shot examples**: [...]

**Previous actions**: [...]

**Current observation**: [...]

Figure 4: An overview of prompts for text-based agents, with abbreviated relevant information as [...]. The complete prompt is at Appendix C.2.

To facilitate the interactions of LLM and MLLM agents with an Android emulator, we define a screen translator that parses the observation from the Android view hierarchy (Zhang et al., 2023; Yang et al., 2023b). This translator converts screen layout information (i.e., the Android view hierarchy presented in XML format) into a text description of the UI elements. Each description includes a numeric tag and details of each UI element, such as the class information specifying the type of UI. Additionally, we define a set of possible action options that can be selected by agents in text format, as detailed in Table 1.

In prompts, we include the role of agents, action space definition, goal, (optional) few-shot examples, previous actions taken by the agent, and the current observation. Our prompts, outlined in Figure 4, also incorporate the Chain-of-Thought technique (Wei et al., 2022) to enhance the reasoning ability by enforcing a certain output format. We illustrate an overview of LLM agents in Appendix C.1.

## 3.2 OPEN-SOURCE MODELS

Despite the promising results of closed-sourced foundation models, leveraging them presents several challenges such as the difficulties in fine-tuning. In our benchmark analysis, we also explore agents employing customized models, named custom agents, using either a fine-tuned open-source foundation model (i.e., Llama-3) or modules trained from scratch. These custom agents process task instructions and screen layout descriptions in text form to produce text-based actions, similar to agents using closed-source LLMs or MLLMs. We fine-tune these models based on actions collected from human experts, with prompts that specify their general roles and relevant task-specific information.[1]

The custom agents equipped with an encoder using a vision and language model (VLM encoder), with the policy denoted by $\pi_\theta$, take task instructions $c$ and screen images $o_t$ as inputs, and output discrete actions $a_t$. Input embeddings, extracted using a pre-trained VLM encoder (Yang et al., 2023a), are processed through a transformer module (Vaswani et al., 2017) to generate actions. Each action is a vector of size 385, where the first 378 values correspond to tapping pre-defined locations (14×27 bins) on the screen, four values to swiping directions (up, down, right, left), and the last three values to pressing buttons (back, home, overview). For further details on the network architecture, we refer readers to Appendix C.3. We train custom agents with VLM encoder from scratch using BC or RL. For BC, agents are optimized to imitate the human expert demonstrations $\mathcal{D} = \{(o_t, a_t^*, c)\}$ by minimizing the cross entropy loss $L_{\text{BC}}$:

$$\sum_{(o_t, a_t^*, c) \sim \mathcal{D}} L_{\text{BC}}(\pi_\theta(a_t | o_t, c), a_t^*).$$

We provide a more detailed explanation of the algorithmic designs for RL training in Appendix E.1.

## 4 EXPERIMENTS

We design our experiments to investigate the following research questions:

---

[1]We do not employ the CoT technique in custom agents due to challenges in preparing the dataset with such thought processes.

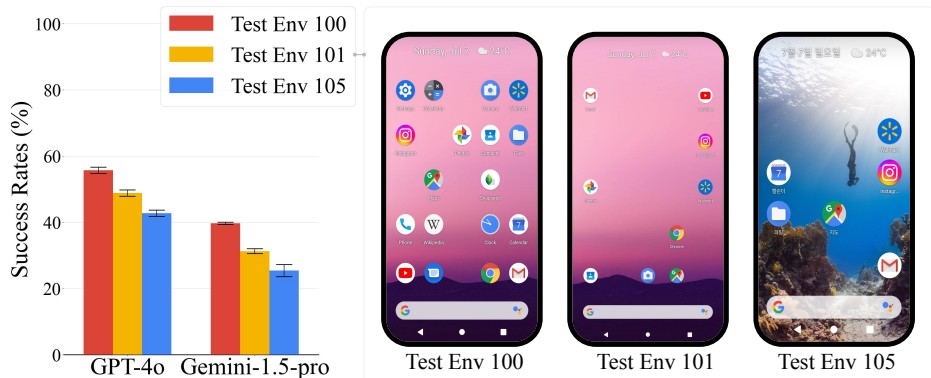

Figure 5: Average success rates of state-of-the-art LLM agents on all 131 tasks in three test environments. We report the zero-shot performance of the agents with three runs. The different success rates among environments reveal the unique challenge of B-MoCA in assessing the generalization ability.

- How well do agents employing state-of-the-art LLMs perform daily tasks in B-MoCA? (Figure 5)
- What are the different behaviors of the baseline agents? (Figure 6)
- How are the LLM agents affected by the randomization of each environmental factor? (Figure 7)
- How do different choices for employing LLMs affect performance? (Table 2)
- How crucial is data diversity when training custom agents? (Figure 8)

## 4.1 EXPERIMENTAL SETUP

We employ three state-of-the-art foundation models. For LLM agents, we employ closed-source LLMs: GPT-4o (`gpt-4o-2024-05-13`; OpenAI 2024) and Gemini-1.5-pro (`gemini-1.5-pro-001`; Gemini et al. 2023 with text-only input. We also consider an open-source model: Llama-3 (`meta-llama/Meta-Llama-3-70B-Instruct`; Meta 2024). We evaluate LLM agents with and without few-shot examples. For few-shot learning, we select examples from a pool of 210 human expert demonstrations across 35 training environments (see Appendix D.1 for dataset collection). For MLLM agents, we leverage GPT-4o and Gemini-1.5-pro by providing additional image inputs. We provide more details on the configurations in Appendix D.2.

We experiment with two types of custom agents.[2] For the custom agents using open-source LLM, we fine-tune Llama-3 (`meta-llama/Meta-Llama-3-8B-Instruct`) using human demonstrations from 35 training environments, adopting LoRA (Hu et al., 2022). For custom agents using VLM encoders, we fully fine-tune the cross-attention transformer module and visual encoder. We refer the readers to Appendix D.3 for details on the training procedure. We also provide experimental results with agents trained using RL in Appendix E.

We conduct two main experiments. In the first main experiment, we examine state-of-the-art closed-source LLM agents (i.e., GPT-4o and Gemini-1.5-pro) on all 131 tasks. The agents are evaluated in zero-shot on three test environments: a vanilla environment with all target applications in the home screen ("Test Env 100" with id 100, described in Appendix A.2), another environment with randomized settings of icons in the home screen and size randomized ("Test Env 101" with id 101), and the other environment with randomized settings of icons in the home screen, size, wallpaper, and language ("Test Env 105" with id 105). These varying configurations challenge the generalization ability of agents. For example, in tasks for setting the alarm, the clock UI appears to be either circular in "Test Env 100" or rectangular in Test Env 105".

In the second main experiment, we study all the baseline agents on six representative challenging tasks: `Alarm(simple)`, `Alarm(complex)`, `Calculator`, `Call`, `Language`, and `Wikipedia`.

---

[2]We also have conducted a study with the agents trained with RL, and the relevant report is available at the Appendix E

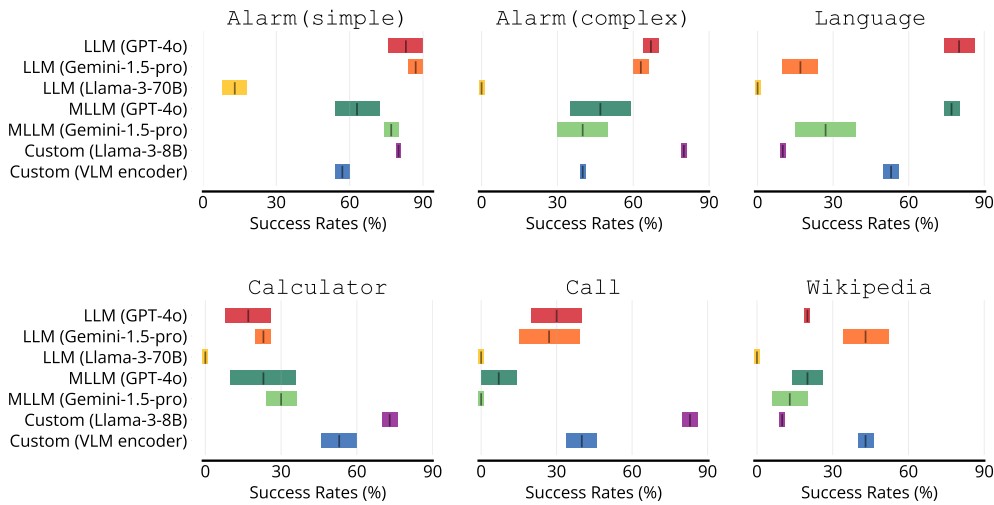

Figure 6: Average success rates of baseline agents in the test environments. We report the mean and standard error across three runs. LLM agents and MLLM agents are evaluated using few-shot examples. Custom agents are trained with BC using human demonstrations.

These tasks are selected as they require navigating multiple pages and manipulating diverse UI elements in a long horizon (i.e., episode length of at least 7). In the `Alarm` tasks, for example, the agents not only need to reach the alarm tab in the clock application but also operate the clock UI to set the alarms. We provide exemplary expert demonstrations for these tasks in Appendix B.2. For each task, the goal instruction provided to the agents is as follows:

- `Alarm(simple)`: "create alarm at 10:30 am"
- `Alarm(complex)`: "create alarm at 10:30 am on every weekday"
- `Language`: "go to the 'add a language' page in setting"
- `Calculator`: "input 'cos(180)' in Calculator"
- `Call`: "call the white house (202-456-1111)"
- `Wikipedia`: "disable the top 2 and 'randomizer' topics at feed customize setting on Wikipedia and go back to the feed"

For all experiments, we report the mean and standard error across three different runs.

## 4.2 MAIN RESULTS

Figure 5 shows the success rates of LLM agents on all 131 tasks, and Figure 6 displays the success rates of LLM agents, MLLM agents, and custom agents on the six challenging representative tasks. The agents employing the foundation model complete simple tasks with high performance by leveraging their pre-trained base knowledge. For example, LLM agents using GPT-4o achieve success rates higher than 80% on the `Alarm(simple)` task. However, their performances significantly decrease as the tasks become complex (e.g., significant degradation on the `Alarm(complex)` task). On the other hand, custom agents using policies trained from scratch imitate the behaviors of experts and exhibit average success rates greater than or equal to 40% on all six challenging representative tasks. However, these agents still show a lack of generalization ability (less than 60%) on these tasks. Similarly, custom agents using fine-tuned Llama-3 show an extreme discrepancy of proficiencies across the six challenging tasks. The shortcomings of these baselines call for new algorithms for building mobile device control agents.

We provide more remarks on each agent type below.

**Challenges for agents using ready-made foundation models**   The LLM agents show remarkable performances in controlling mobile devices, even in zero-shot as shown in Figure 5. However, we

witness several limitations of agents employing the foundation models, across both LLM agents and MLLM agents. First, LLM and MLLM agents frequently hallucinate task completion. For example, on the `Calculator` task, the agents often conclude that a task is complete before entering all the requested strings (e.g., entering 'cos(18)' instead of 'cos(180)'). Second, these agents face difficulties with long-horizon tasks that require multiple interactions. For instance, they frequently make mistakes when typing the sequence of numbers on the `Call` task.

**Effects of environment randomization on LLM agents**    The success rates of LLM agents decrease as the environments are randomized, as displayed in Figure 5. In the "Test Env 101" environment, we find that both LLM agents using GPT-4o and Gemini-1.5-pro make simple mistakes when the icon locations are randomized. For example, the agent employing GPT-4o taps the Walmart icon to open the Wikipedia application, if the Wikipedia application is not available on the home screen. In the "Test Env 105" environment, we observe more performance degradation. To be specific, the agents suffer from handling three main randomizations: fewer icons on the home screen, understanding the text descriptions of UI elements in Korean, and a larger icon size setting. We include a more rigorous analysis of the effect of each environmental feature in Section 4.3.

**Comaprison between LLM agents and MLLM agents**    In our experiments, gains from additional image inputs are marginal or even detrimental, as shown by comparing performances with and without additional screenshot inputs (see red vs. dark green and orange vs. light green). We expect that this is due to domain gaps in visual inputs. Additionally, we hypothesize that multi-modal agents suffer from the increased length of input sequence associated with additional image tokens. A similar observation was made in recent work involving agents that control desktop computers (Xie et al., 2024). These results indicate the remaining headroom in leveraging multi-modal inputs more effectively.

**Differences between closed-source and open-source LLMs**    In the setting of using the ready-made foundation models, closed-source LLMs like GPT-4o (red) outperform open-sourced LLMs like Llama-3 (yellow) across all six representative challenging tasks, as shown in Figure 6. Notably, the ability to understand current screen layouts significantly differs between them. For example, when we ask LLMs to describe what they observe in the input, agents using closed-sourced LLMs like GPT-4o typically produce more detailed texts, such as a precise list of application icons. Furthermore, LLM agents using GPT-4o exhibit better planning abilities. For instance, when the target application is not visible on the current home screen, LLM agents using GPT-4o often swipe the screen to navigate or explore the app list menu. In contrast, LLM agents using Llama-3 tend to tap irrelevant applications visible in the current view. The results of evaluations across all 131 tasks are included in Appendix D.4.

**On custom agents using fine-tuned LLMs**    We observe that custom agents leveraging fine-tuned Llama-3 models achieve superior performance on several complex tasks, exhibiting greater proficiency compared to agents using standard Llama-3 across all tasks. These agents successfully complete the `Alarm` tasks (both simple and complex), on the test environments demonstrating high similarities with the training environments. Their proficiencies in many tasks reveal the benefits of fine-tuning LLMs to build agents. However, their failure on the `Language` and `Wikipedia` tasks in most of the test environments demonstrates a clear limitation, especially on the generalization ability.

**Generalization ability of custom agents**    In our experiments, the custom agents achieve reasonable performances in many complex tasks, even where agents using closed-source foundation models fail. The custom agents successfully imitate expert behaviors in navigating applications and manipulating diverse UI elements in training environments. However, they still show limitations on generalization ability. The agents using fine-tuned Llama-3, for example, fail to generalize their behaviors to environments having a language setting of Korean and a device setting of Tablet. Also, the agents equipped with VLM encoder largely degrade in test environments, while achieving high success rates in training environments (e.g., higher than 90%; see Appendix D.5 for training performances). Specifically, they struggle with tasks involving severe visual changes induced by unseen device configurations. These observations highlight the need to develop more efficient algorithms that improve generalization against changes in device configurations.

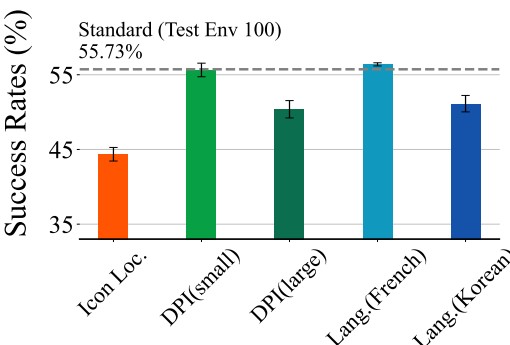 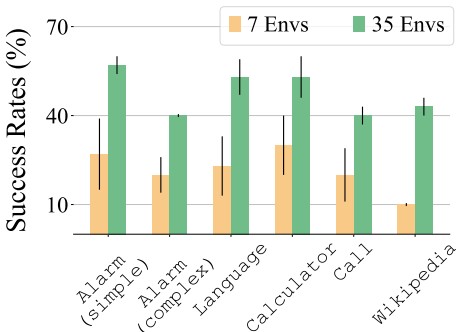

Figure 7: The impacts of each randomized environmental feature. We report the success rates of LLM agents using GPT-4o in all tasks. Icon location affects the agents most significantly.

Figure 8: Success rates of agents trained with BC on varying numbers of training environments. A general trend of escalating success rates with more environments appears.

### 4.3 FURTHER ANALYSES

In this section, we further analyze the experiments conducted in B-MoCA.

**Generalization ability of LLM agents on each environmental factor**  To examine the impact of randomizing each environment feature, we design five new environments by modifying individual components of a standard test environment (referred to as 'Standard'). These modifications include changing the icon location on the home screen, adjusting the DPI (options of small or large), and changing the language settings (French or Korean), while keeping all other settings constant. As shown in Figure 7, the impact of each environmental feature varies. LLM agents using GPT-4o agents (text-only without few-shot examples) are robust to language changes. However, they exhibit significant performance decreases when icon locations are altered, requiring them to explore in order to locate target applications.

**LLM agents with few-shot examples**  Table 2 compares the performance of LLM agents using GPT-4o with and without few-shot examples. We observe that providing three examples to the agents improves the success rates on several tasks (e.g., `Alarm(complex)` and `Call`). However, few-shot examples do not always enhance performance and can sometimes even be detrimental. Specifically, the agents often struggle to utilize these examples effectively, becoming confused by the exemplary observations and choosing actions from the human expert examples that are irrelevant to the current context. For example, they frequently select the final action in the exemplary trajectory, regardless of their actual position within a different screen. This highlights a significant challenge in using few-shot examples for LLM agents: naively mimicking actions from human demonstrations often leads to errors, particularly when the UI elements vary across different device settings. We also provide the analysis of MLLM agents in Appendix D.6, where the overall trend is similar.

|  | LLM agents (zero-shot) | LLM agents (few-shot) |
|---|---|---|
| Alarm(simple) | $87 \pm 09$ | $83 \pm 07$ |
| Alarm(complex) | $33 \pm 07$ | $67 \pm 03$ |
| Language | $77 \pm 03$ | $80 \pm 06$ |
| Calculator | $17 \pm 03$ | $17 \pm 09$ |
| Call | $00 \pm 00$ | $30 \pm 10$ |
| Wikipedia | $30 \pm 06$ | $20 \pm 00$ |
| Average | $41 \pm 04$ | $49 \pm 05$ |

Table 2: Success rates of LLM agents using GPT-4o with and without few-shot examples. While including few-shot examples enhances performance on the `Language` task, it significantly reduces performance on the `Calculator` task.

**Effect of training data diversity on custom agents**  We train custom agents with varying numbers of training environments (see Appendix D.1 for more details of the experimental setup). As shown in Figure 8, the performance of custom agents escalates as the number of training environments increases. Specifically, on the `Language` task, the agents exhibit success rates of 23% and 53% as trained with 7 and 35 training environments, respectively. We believe this verifies the efficacy of the environment randomization incorporated in our benchmark for developing practical agents.

## 5 RELATED WORK

**Foundation models for decision-making system**  Inspired by the strong emergent properties of foundation models (Brown et al., 2020; Wei et al., 2022), many researches have adopted LLMs to develop decision-making system (Yao et al., 2023; Shinn et al., 2023). In robot learning, for example, LLMs have been widely equipped for reasoning, planning, manipulation, and navigation (Driess et al., 2023; Liang et al., 2023; Huang et al., 2023). Furthermore, agents with LLMs have shown capabilities of performing interesting tasks in numerous simulated worlds, including game environments (Wang et al., 2023; Tan et al., 2024) and virtual reality (Qian et al., 2023; Yang et al., 2024). In recent days, focusing on practicalness, solving computer tasks with foundation models has also been actively explored (Nakano et al., 2021; Furuta et al., 2023). We further study the abilities of foundation models to control mobile devices toward assistive agents in real life.

**Developing assistive agent for device control**  For agents that effectively understand and manipulate the UI elements, a large body of work has leveraged structural information, such as document object model in HTML or Android view hierarchy (Branavan et al., 2010; Gur et al., 2019). In addition, methods for equipping agents with the ability to understand information-rich screen images have been widely investigated, mainly with vision-based reinforcement learning (Liu et al., 2018; Humphreys et al., 2022; Shaw et al., 2023). Recently, diverse strategies to build device control agents with foundation models are introduced, including prompting methods (Wen et al., 2023; Kim et al., 2023), instruction-tuning (Furuta et al., 2023), fine-tuning with images (Zhan & Zhang, 2023; Hong et al., 2023), and visual prompting (Yan et al., 2023; Yang et al., 2023b). Here, we present an elaborate analysis of the main methods for building mobile device control agents.

**Benchmark for decision-making agents**  There have been continuous efforts to build reliable benchmarks for sequential decision-making in video games (Bellemare et al., 2013), locomotion (Brockman et al., 2016), and robotic manipulation (James et al., 2020). Lately, researchers have proposed benchmarks for solving device control tasks, viewing it as another decision-making problem. For example, Yao et al. (2022) and Zhou et al. (2024) have presented benchmark simulating web platforms, while Toyama et al. (2021), Shvo et al. (2021), and Zhang et al. (2023) have suggested RL environments adopting Android emulators. In this work, inspired by special-purpose benchmarks quantifying the generalization ability of the agents (Cobbe et al., 2020; Stone et al., 2021), we newly propose a benchmark with a randomization feature.

## 6 CONCLUSION

We present B-MoCA, a new benchmark designed for evaluating mobile device control agents. Our benchmark provides diverse tasks applicable to everyday routines and environments that simulate numerous device configurations. We conduct extensive experiments and demonstrate that B-MoCA can serve as a standardized platform for developing different types of agents in a unified setting. Finally, we mention several limitations and promising future directions of this work:

- *Tasks with text typing:* While agents can input text by touching the soft keyboard on the screen, it demands excessively long interactions. We plan to include tasks requiring text typing, such as web search or e-mail sending, with advanced interfaces in the future.

- *Open-ended tasks and reward modeling:* Since the ADB-based success detector does not capture the semantics of agent behaviors, tasks with ambiguous success criteria are hard to evaluate. Alternatively, we believe employing the reward model learned from demonstrations (Fan et al., 2022) can be used for integrating open-ended tasks.

- *More on agents with foundation models:* LLMs can be employed in different ways, such as using them as a high-level planner to operate a set of pre-defined APIs (Chen & Li, 2024) or neural network policies (Ahn et al., 2022) as low-level actors. Also, we note that fine-tuning LLMs is highly promising, as training custom agents with demonstrations results in compatible performances, leaving it as future work.

Toward practical mobile device control agents, we hope that B-MoCA stands as a valuable platform with helpful resources for innovative breakthroughs.

ETHICS STATEMENT

This study proposes a benchmark designed to assess interactive mobile device management agents, with social opportunities to enhance user accessibility and aid those facing disabilities. We caution users about privacy concerns, while we try to eliminate such potentials during task designs. Noting the importance of research for preventing malicious usages of device control agents, we emphasize B-MoCA as a useful test bed.

REPRODUCIBILITY STATEMENT

We ensure the reproducibility of our results by providing comprehensive details about our benchmark. Additionally, we include code materials in the supplementary materials which can be valuable for reproducing our environments and experiments.

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

# Appendix:

## Benchmarking Mobile Device Control Agent across Diverse Configurations

## A  ENVIRONMENT DETAILS

### A.1  ENVIRONMENT IMPLEMENTATION AND INTERFACE

**Environment**  B-MoCA is based on Android OS for real-system interactive evaluation. The environment is simulated with Android virtual devices, containing the device hardware profile, system image, storage area, and other relevant properties. The dynamics of the environments, such as the transition rules, are governed by Android OS and applications.

Each environment is represented as an Android device, running on top of the Android emulator. To be specific, we define each environment as a snapshot, a stored image of the Android virtual device. Each snapshot is built by saving an image of the target device after the configurations. These configurations include installing third-party applications and randomizing the features of environments. The features of the environments we randomize encompass placing icons in random locations, setting dots per inch (DPI), modifying wallpapers, and changing the language. During the configuration, adjusting several device settings for accurate evaluation, such as changing the database of applications, is also conducted.

To facilitate interactions between the environment and agents, we develop a set of interfaces on top of the Python library AndroidEnv (Toyama et al., 2021). The interfaces we develop encompass various functionalities: to provide the task descriptions in text to the agents, to capture screenshots of the virtual device, to provide the Android view hierarchy in XML format and parse the text description of the screen, to extract dual-gesture actions from text-based actions, and to deliver the dual-gesture action to the Android emulator.

**Interaction frequency**  The Android emulators run asynchronously, independent of the agent that interacts with the environments. However, this asynchronicity between the agent and the environment may cause several issues such as incomplete transition of the environments or delayed success signals. To alleviate the issue, we adjust the interaction frequency between agents and environments. Specifically, this adjustment is operated by forcing the agent to wait a pre-defined time before fetching the screen information from the environment. In our experiments, we fix the interaction frequency during evaluation to be 1/3Hz across all types of agents, except Instagram with longer latency (e.g., 1/30Hz) as locale change requires additional loading. We also allow users to adjust this interaction frequency.

**Observation space**  The observation space is comprised of either a screen image, a text description of the screen in XML formats based on the Android view hierarchy, or both. The screen images are used for multi-modal large language model (MLLM) agents and custom agents. Each image is resized into a resolution of $256 \times 512$ for MLLM agents and $128 \times 256$ for custom agents.

The text descriptions are used for agents with LLMs and MLLMs. To build the text description, the Android debug bridge (ADB) UI Automator is employed for acquiring the Android view hierarchy in XML format. A pre-defined screen translator, then, converts the information of UI elements in the XML file into a set of text descriptions of UI elements. The description includes a numeric tag and details of the UI elements, including the class name or content descriptions. Additionally, we provide attributes if the UI elements are checked or selected and (optionally) the bounding box x-y coordinates specifying the location of the elements, such as the slider interface. In our interface, the parser captures the descriptions of all the nodes in the Android view hierarchy, not specified to the leaf nodes.

**Action space**  The action space of the agents is defined as either continuous or discrete actions. The continuous action, here, refers to a dual-gesture action $\{a \mid a = (y_{\text{touch}}, x_{\text{touch}}, y_{\text{lift}}, x_{\text{lift}}) \in \mathbb{R}^4\}$, similar to Rawles et al. (2023). Each value of dual-gesture action $a$ is normalized to be in between $[-1, 1]$ with respect to the screen resolutions. The former two values specify the location of the screen to touch, while the latter two values determine the location of the screen to lift. This definition

enables interpreting useful actions in digital device control, i.e., tapping or swiping the screens, in a precise and compressive manner. Also, our interface allows pressing the navigator buttons available by touching the screen to support the essential actions for manipulating Android devices.

The discrete action, on the other hand, refers to the pre-defined set of actions for tapping, swiping, and button-pressing. For agents generating text-based actions, the discrete action is defined to be each action option (i.e., callable function). For agents trained from scratch, we define each action to be a vector with the size of 378, where the first 378 values correspond to tapping pre-defined locations (14×27 bins) on the screen, four values to swiping directions (up, down, right, left), and the last three values to pressing buttons (back, home, overview).

For dual-gesture actions, we implement an interface that determines whether the action is a tap, swipe, or pressing of navigation buttons i.e., back, home, and overview. The action parsing interface converts the action into taps, swipes, or pressing buttons following the rule as follows:

- The action is `tapping`, if $d((x_{\text{touch}}, y_{\text{touch}}), (x_{\text{lift}}, y_{\text{lift}})) < \text{threshold}$
  - The `tapping` is to press BACK button, if $(x_{\text{touch}}, y_{\text{touch}}) = (0.95, 0.22)$
  - The `tapping` is to press HOME button, if $(x_{\text{touch}}, y_{\text{touch}}) = (0.95, 0.50)$
  - The `tapping` is to press OVERVIEW button, if $(x_{\text{touch}}, y_{\text{touch}}) = (0.95, 0.78)$
- The action is `swiping`, if $d((x_{\text{touch}}, y_{\text{touch}}), (x_{\text{lift}}, y_{\text{lift}})) \geq \text{threshold}$,

where the threshold value is defined as 0.14. This value is adjustable by users, while we find that the value of 0.14 ensures proper interactions over UI elements, e.g., `tapping` the target application icon, in all of our experiments. These specific values are tested to be consistent across different device types, ensuring that the positions correspond to the correct buttons in all B-MoCA environments.

For agents with foundation models, we further define action converter that translate text-based actions into legal emulator actions. Following the action space definition, the action options are designed to be either dual-gesture actions or discrete actions. We prompt the LLM agents to output actions among six possible options: dual-gesture action, tap, swipe, press("HOME"), press("BACK"), and press("OVERVIEW"). The action converter translate the text-based actions into the legal actions as below:

- For the dual-gesture action, it converts the text action into the four floating points by rounding each value into the second decimal point.
- For tap actions, the agent outputs an integer value specifying the numeric tag assigned to the UI element. Given the tapping action with a numeric tag, it converts the action into a tapping dual-gesture action with the bounding box information of the chosen UI element.
- For swipe actions, a direction 'up', 'down', 'left' and 'right' is converted into a corresponding dual-gesture action with the value of $(0.8, 0.5, 0.2, 0.5)$, $(0.2, 0.5, 0.8, 0.5)$, $(0.5, 0.2, 0.5, 0.8)$, and $(0.5, 0.8, 0.5, 0.2)$, respectively.
- For the action press("HOME"), press("BACK"), and press("OVERVIEW"), it converts the actions to tap the corresponding screen location.

During the evaluation, we ignore the action in the wrong format by skipping the transition of the environments but penalizing the agents by incrementing the steps taken.

### A.2 TRAINING AND TEST ENVIRONMENTS CONFIGURATIONS

We construct 45 unique environments in B-MoCA, where 35 environments are for training and 10 environments are for testing. Each environment is provided with a unique identification (ID) number, to distinguish the environments easily. From Table 3 to Table 5 show the list of the device configurations and the home screen images of exemplary environments.

To construct environments, we use popular device types: Pixel 3, Pixel 4, Pixel 4 XL, Pixel 6, and WGXA Tablet. For training environments, only Pixel 3 is employed. For evaluation environments, we use all device types Pixel 3, Pixel 4, Pixel 4 XL, Pixel 6, and WGXA Tablet. In these models, we alter the icon and font sizes by changing the dots per inch (DPI) values of the devices. For each device type, we prepare three different sizes that users can select. We, then, change the wallpaper

with 13 images collected from a free license image website. These wallpaper image files are shared in the open-source repository. We also customize the background images with the dark theme mode. If the dark theme mode is activated, the device provides screen images with light-dark color reversed. For instance, the wallpaper of the application list page is white in the default setting, while it becomes black with dark theme mode activated. Furthermore, we incorporate changes in locale, specifying the language and location of the devices. 12 different locales are used for 35 training environments, while we include three more locales for the test environments.

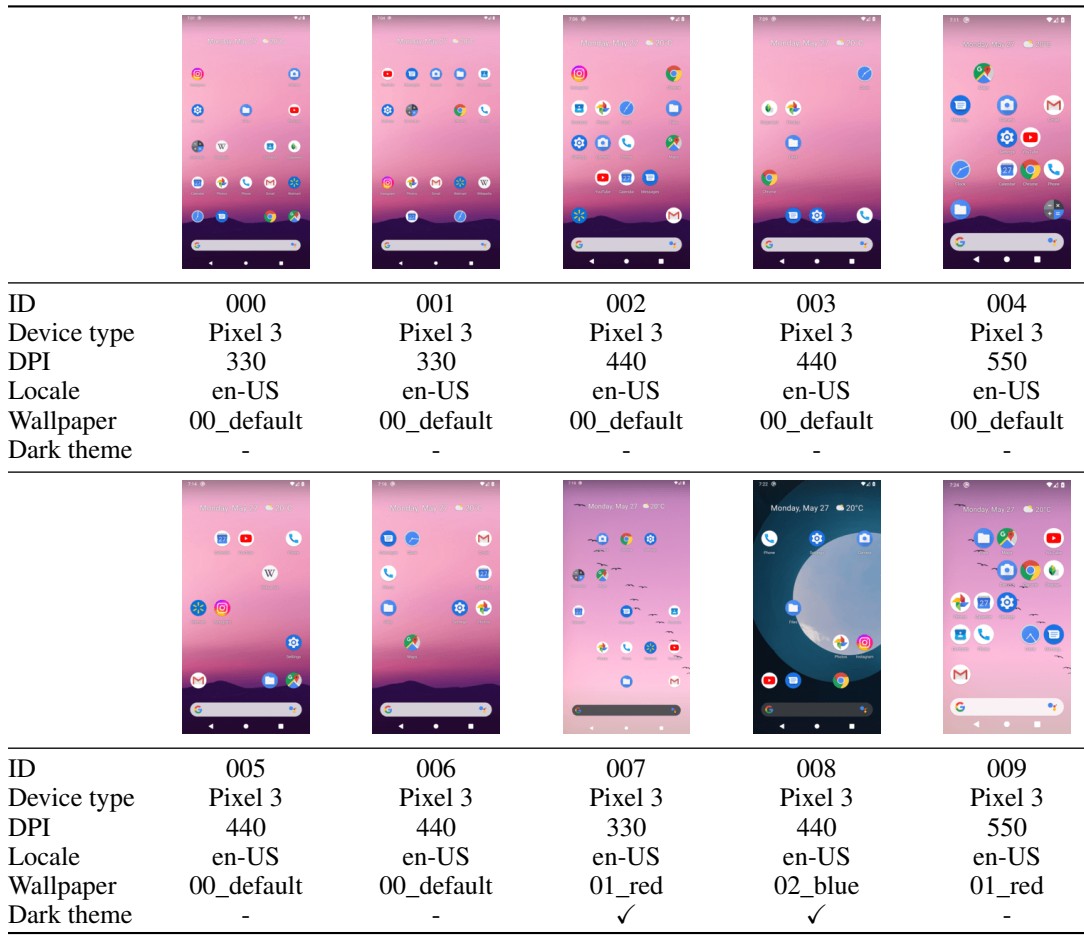

| ID | 000 | 001 | 002 | 003 | 004 |
|---|---|---|---|---|---|
| Device type | Pixel 3 | Pixel 3 | Pixel 3 | Pixel 3 | Pixel 3 |
| DPI | 330 | 330 | 440 | 440 | 550 |
| Locale | en-US | en-US | en-US | en-US | en-US |
| Wallpaper | 00_default | 00_default | 00_default | 00_default | 00_default |
| Dark theme | - | - | - | - | - |

| ID | 005 | 006 | 007 | 008 | 009 |
|---|---|---|---|---|---|
| Device type | Pixel 3 | Pixel 3 | Pixel 3 | Pixel 3 | Pixel 3 |
| DPI | 440 | 440 | 330 | 440 | 550 |
| Locale | en-US | en-US | en-US | en-US | en-US |
| Wallpaper | 00_default | 00_default | 01_red | 02_blue | 01_red |
| Dark theme | - | - | ✓ | ✓ | - |

Table 3: The device configuration of each environment with the home screen image.

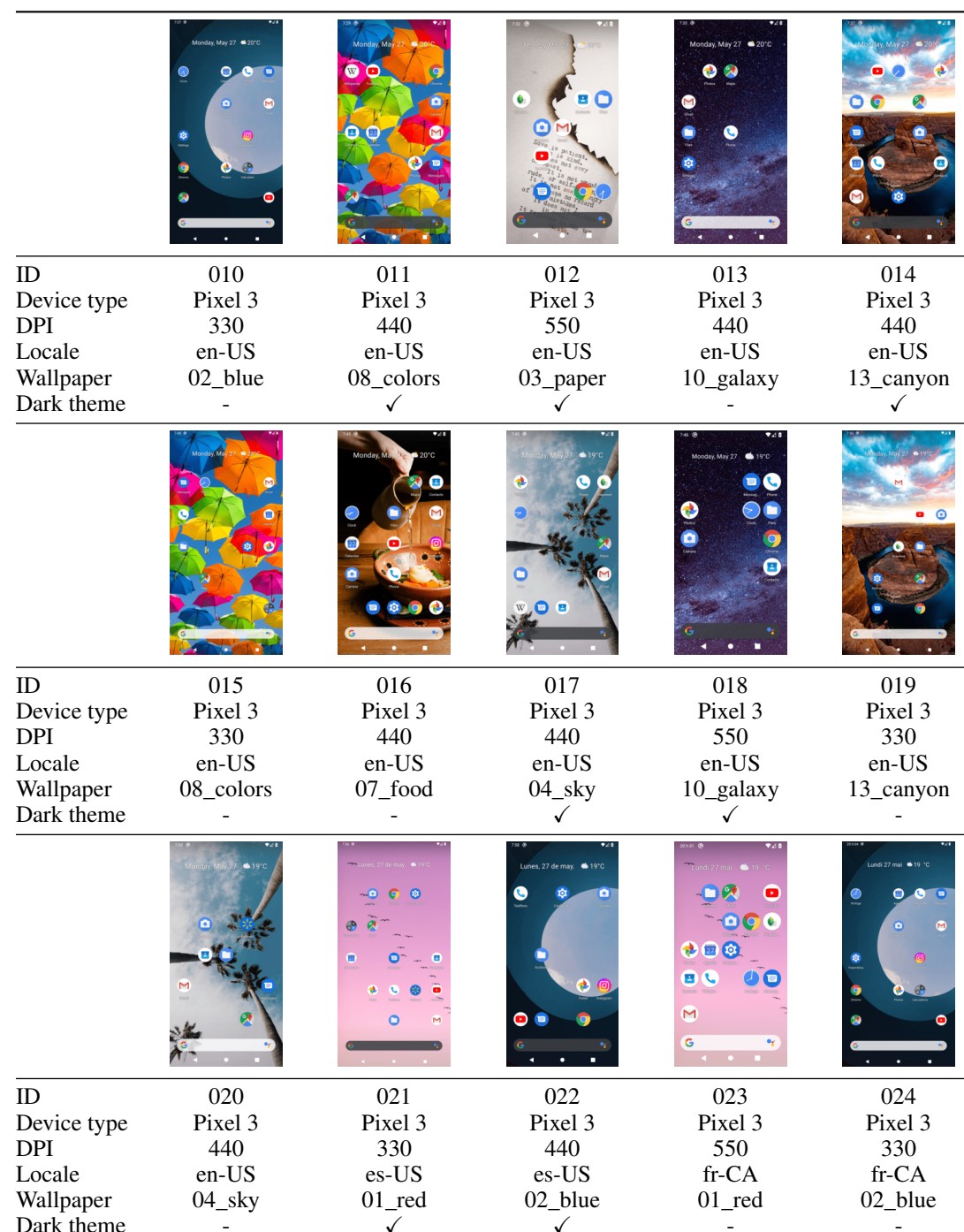

| ID | 010 | 011 | 012 | 013 | 014 |
|---|---|---|---|---|---|
| Device type | Pixel 3 | Pixel 3 | Pixel 3 | Pixel 3 | Pixel 3 |
| DPI | 330 | 440 | 550 | 440 | 440 |
| Locale | en-US | en-US | en-US | en-US | en-US |
| Wallpaper | 02_blue | 08_colors | 03_paper | 10_galaxy | 13_canyon |
| Dark theme | - | ✓ | ✓ | - | ✓ |

| ID | 015 | 016 | 017 | 018 | 019 |
|---|---|---|---|---|---|
| Device type | Pixel 3 | Pixel 3 | Pixel 3 | Pixel 3 | Pixel 3 |
| DPI | 330 | 440 | 440 | 550 | 330 |
| Locale | en-US | en-US | en-US | en-US | en-US |
| Wallpaper | 08_colors | 07_food | 04_sky | 10_galaxy | 13_canyon |
| Dark theme | - | - | ✓ | ✓ | - |

| ID | 020 | 021 | 022 | 023 | 024 |
|---|---|---|---|---|---|
| Device type | Pixel 3 | Pixel 3 | Pixel 3 | Pixel 3 | Pixel 3 |
| DPI | 440 | 330 | 440 | 550 | 330 |
| Locale | en-US | es-US | es-US | fr-CA | fr-CA |
| Wallpaper | 04_sky | 01_red | 02_blue | 01_red | 02_blue |
| Dark theme | - | ✓ | ✓ | - | - |

Table 4: The device configuration of each environment with the home screen image.

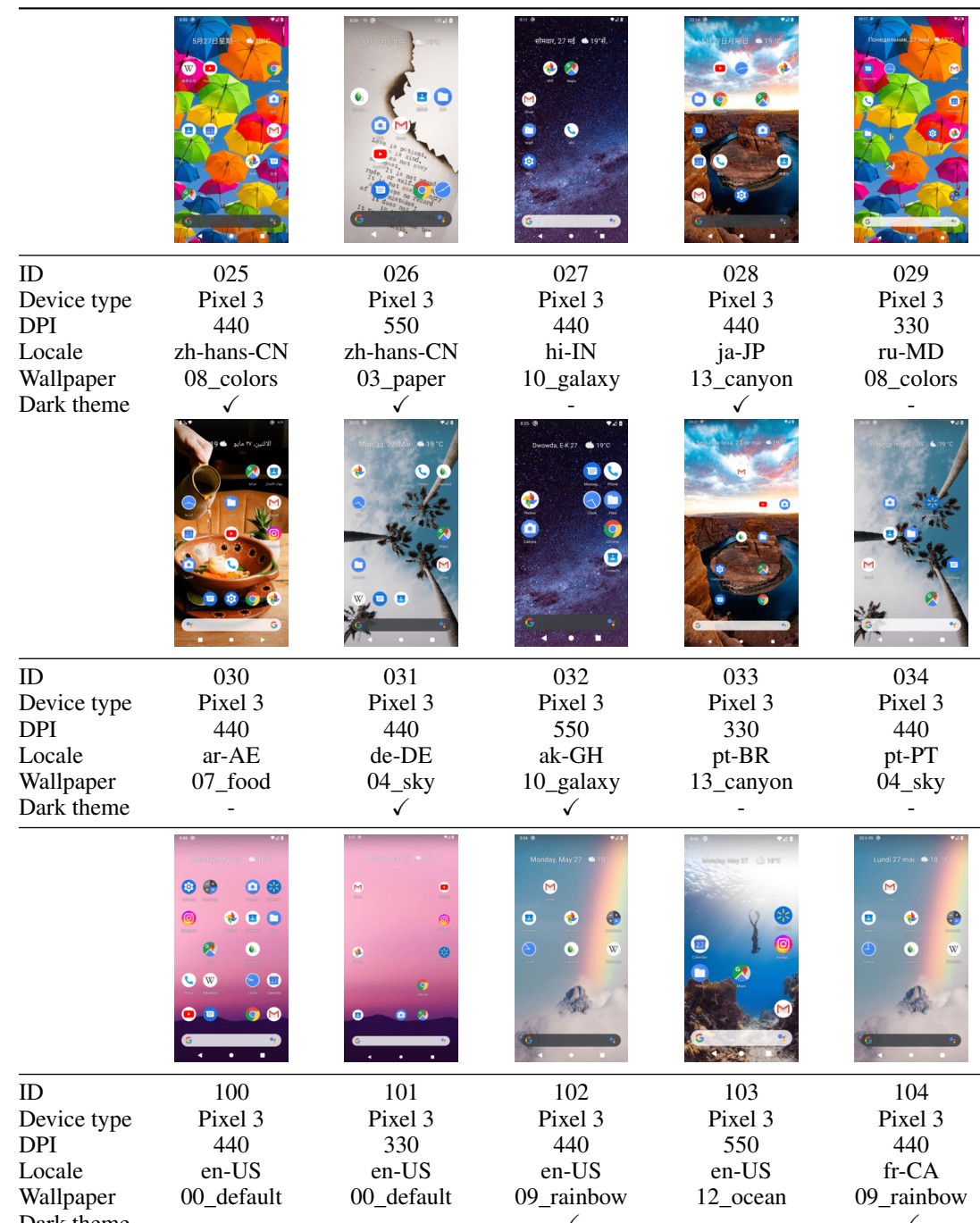

| ID | 025 | 026 | 027 | 028 | 029 |
|---|---|---|---|---|---|
| Device type | Pixel 3 | Pixel 3 | Pixel 3 | Pixel 3 | Pixel 3 |
| DPI | 440 | 550 | 440 | 440 | 330 |
| Locale | zh-hans-CN | zh-hans-CN | hi-IN | ja-JP | ru-MD |
| Wallpaper | 08_colors | 03_paper | 10_galaxy | 13_canyon | 08_colors |
| Dark theme | ✓ | ✓ | - | ✓ | - |
| ID | 030 | 031 | 032 | 033 | 034 |
| Device type | Pixel 3 | Pixel 3 | Pixel 3 | Pixel 3 | Pixel 3 |
| DPI | 440 | 440 | 550 | 330 | 440 |
| Locale | ar-AE | de-DE | ak-GH | pt-BR | pt-PT |
| Wallpaper | 07_food | 04_sky | 10_galaxy | 13_canyon | 04_sky |
| Dark theme | - | ✓ | ✓ | - | - |
| ID | 100 | 101 | 102 | 103 | 104 |
| Device type | Pixel 3 | Pixel 3 | Pixel 3 | Pixel 3 | Pixel 3 |
| DPI | 440 | 330 | 440 | 550 | 440 |
| Locale | en-US | en-US | en-US | en-US | fr-CA |
| Wallpaper | 00_default | 00_default | 09_rainbow | 12_ocean | 09_rainbow |
| Dark theme | - | - | ✓ | - | ✓ |

Table 5: The device configuration of each environment with the home screen image.

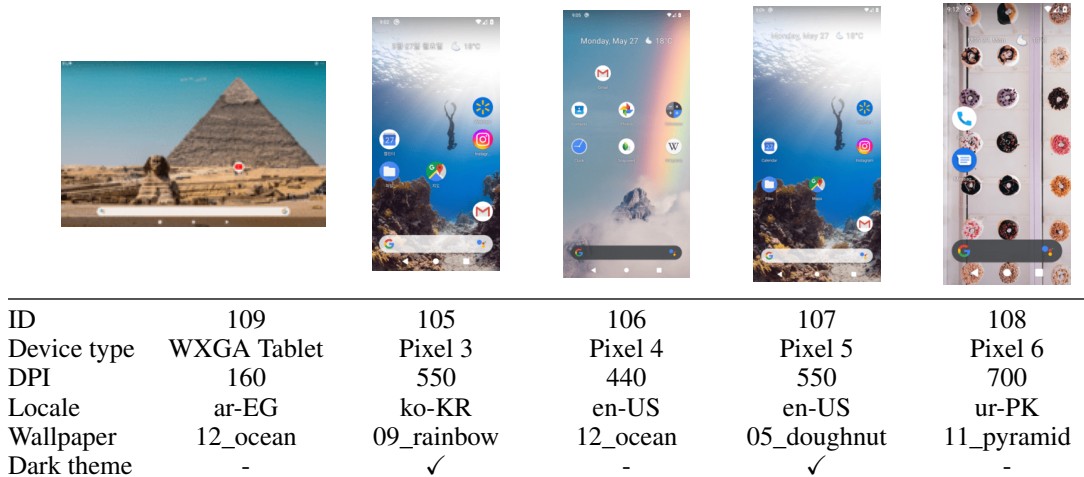

| ID | 109 | 105 | 106 | 107 | 108 |
|---|---|---|---|---|---|
| Device type | WXGA Tablet | Pixel 3 | Pixel 4 | Pixel 5 | Pixel 6 |
| DPI | 160 | 550 | 440 | 550 | 700 |
| Locale | ar-EG | ko-KR | en-US | en-US | ur-PK |
| Wallpaper | 12_ocean | 09_rainbow | 12_ocean | 05_doughnut | 11_pyramid |
| Dark theme | - | ✓ | - | ✓ | - |

Table 6: The device configuration of each environment with the home screen image.

## B    TASK DETAILS

### B.1    LIST OF DAILY TASKS

B-MoCA presents 131 daily tasks that are common in everyday life. The tasks are designed to operate in diverse environments seamlessly and cover commonly used applications. Daily tasks effectively simulate a wide range of essential skills for mobile device control problems, such as manipulating UI elements (including application icons, checkboxes, and sliders), and can be employed for evaluating mobile device control agents' capabilities in performing tasks that mirror our daily activities. We have defined success criteria for each task by analyzing significant changes in the environment (i.e., three key information sources) observed during task completion in human expert demonstrations. We also set the maximum step limits, which are set for the rigorous evaluation of the agents' proficiency in each task. From Table 7 to Table 9, we include the detailed list of tasks with the detailed success criteria.

The source of success criteria are threefold: the system information, app data, and attributes of the UI elements in the Android view hierarchy. System information includes system log and system setting. For the system log, the success detector checks the system log matches with the criteria defined in a regular expression. The log that matches the criteria should satisfy two parts: a filter specifying the target application or activity (denoted as [...] at the success criteria column in Table 7) and a regex specifying the detail of the log (denoted as "..." at the success criteria column in Table 7). For the system setting, the success detector utilizes ADB commands to retrieve specific information related to fields. The success detector, then, parses the results of these commands with a regular expression to extract the necessary system information and check it (denoted as {"(adb command)": "(regular expression)"} at the success criteria column in Table 7). For attributes of the UI elements, the success detector checks if certain attributes of specified UI elements satisfy the pre-defined conditions. We define the condition with UI elements having certain IDs (denoted as [...] at the success criteria column in Table 7) and the status as the value of attributes (denoted as {"(attribute key)": "(status value)"} at the success criteria column in Table 7). For app data, the success detector checks if the values of particular attributes in either the database or shared preferences (used to store simple data instead of a database as an XML file) meet the criteria. We define the condition as selected attributes from the database or shared preference (path denoted as [...] at the success criteria column in Table 7) to be specific values (denoted as {"(attribute key)": "(status value)"} at the success criteria column in Table 7).

| Application | Task instruction | Step limit | Success criteria (source) | Success criteria (detail) |
|---|---|---|---|---|
| Calculator | "open Calculator" | 4 | UI elements | {"id": "com.google.android.calculator:id/clr", "enabled": "true"} |
| Calculator | "input 1 in Calculator" | 5 | UI elements | {"id": "com.google.android.calculator:id/formula", "text": "1"} |
| Calculator | "input factorial of 6 in Calculator" | 7 | UI elements | {"id": "com.google.android.calculator:id/formula", "text": "6!"} |
| Calculator | "input '1+1' in Calculator" | 8 | UI elements | {"id": "com.google.android.calculator:id/formula", "text": "1+1"} |
| Calculator | "input '3×5' in Calculator" | 8 | UI elements | {"id": "com.google.android.calculator:id/formula", "text": "3×5"} |
| Calculator | "input square root of 25 in Calculator" | 8 | UI elements | {"id": "com.google.android.calculator:id/formula", "text": "$\sqrt{25}$"} |
| Calculator | "input 'cos(60)' in Calculator" | 9 | UI elements | {"id": "com.google.android.calculator:id/formula", "text": "c60"} |
| Calculator | "compute 50% of 28 ('50%28') in Calculator" | 9 | UI elements | {"id": "com.google.android.calculator:id/formula", "text": "50%28"} |
| Calculator | "input '17×23' in Calculator" | 10 | UI elements | {"id": "com.google.android.calculator:id/formula", "text": "17×23"} |
| Calculator | "input '2+24÷3' in Calculator" | 10 | UI elements | {"id": "com.google.android.calculator:id/formula", "text": "2+24÷3"} |
| Calculator | "input 'cos(180)' in Calculator" | 10 | UI elements | {"id": "com.google.android.calculator:id/formula", "text": "c180"} |
| Calculator | "input 'ln(1234)' in Calculator" | 10 | UI elements | {"id": "com.google.android.calculator:id/formula", "text": "l1234"} |
| Calculator | "input the formula for computing sum of the first 5 Fibonacci numbers in Calculator" | 13 | UI elements | {"id": "com.google.android.calculator:id/formula", "text": "0+1+1+2+3" or "1+1+2+3+5"} |
| Calculator | "input the formula for converting 45 degrees to radians ('45×π÷180') in Calculator" | 13 | UI elements | {"id": "com.google.android.calculator:id/formula", "text": "45×π÷180"} |
| Calculator | "input the formula for computing sum of the first 5 prime numbers in Calculator" | 14 | UI elements | {"id": "com.google.android.calculator:id/formula", "text": "2+3+5+7+11"} |
| Calculator | "compute the harmonic mean of 4 and 5 in Calculator" | 15 | UI elements | {"id": ["com.google.android.calculator:id/result_preview", "com.google.android.calculator:id/result_final"], "text": starts with "4.44"} |
| Calculator | "input '5!÷(2!x3!)' in Calculator" | 15 | UI elements | {"id": "com.google.android.calculator:id/formula", "text": "5!÷(2!×3!)"} |
| Calculator | "input '10!÷(2!x8!)' in Calculator" | 15 | UI elements | {"id": "com.google.android.calculator:id/formula", "text": "10!÷(2!×8!)"} |
| Calculator | "compute the geometric mean of 3, 4, and 5 in Calculator" | 18 | UI elements | {"id": ["com.google.android.calculator:id/result_preview", "com.google.android.calculator:id/result_final"], "text": starts with "3.91"} |
| Calendar | "open the calendar app" | 4 | System log | [ActivityTaskManager:I] "^(.*?)START(.*?)com.android.calendar" |
| Camera | "open the camera app" | 4 | System log | [ActivityTaskManager:I] "^(.*?)Start proc(.*?)com.android.camera" |
| Chrome | "open Chrome" | 4 | System log | [ActivityTaskManager:I] "^(.*?)START(.*?)com.google.android.apps.chrome" |
| Chrome | "open a new tab in Chrome" | 5 | UI elements | {"id": "com.android.chrome:id/tab_switcher_button", "content-desc": ".*2.*"} |
| Chrome | "go to search history in Chrome" | 5 | System log | [ActivityTaskManager:I] "^(.*?)START(.*?)chrome.browser.history.HistoryActivity" |

Table 7: Comprehensive list of tasks.

| Application | Task instruction | Step limit | Success criteria (source) | Success criteria (detail) |
|---|---|---|---|---|
| Clock | "open the clock app" | 4 | System log | [ActivityTaskManager:I] "^(.*?)START(.*?)com.android.deskclock" |
| Clock | "turn on alarm at 9 am" | 4 | System log | [AlarmClock:D] "^(.*?)Created new alarm instance" |
| Clock | "go to the stopwatch page in clock" | 5 | System log | [AlarmClock:D] ^(.*?)Events: ［Stopwatch］ ［Show Tab］ ［Tap］ |
| Clock | "go to the alarm page in clock" | 5 | System log | [AlarmClock:D] "^(.*?)Events: ［Alarm］ ［Show Tab］ ［Tap］" |
| Clock | "go to the timer page in clock" | 5 | System log | [AlarmClock:D] "^(.*?)Events: ［Timer］ ［Show Tab］ ［Tap］" |
| Clock | "delete alarm at 9 am" | 5 | System log | [AlarmClock:D] "^(.*?)Removed alarm" |
| Clock | "start the stopwatch in clock" | 5 | System log | [AlarmClock:D] "^(.*?)Start" |
| Clock | "create alarm at 06:30 am" | 11 | System log | [ConditionProviders.SCP:D] "^(.*?)nextUserAlarmTime(.*?)06:30:00" |
| Clock | "create alarm at 10:30 am" | 11 | System log | [ConditionProviders.SCP:D] "^(.*?)nextUserAlarmTime(.*?)10:30:00" |
| Clock | "create alarm at 13:30 pm" | 11 | System log | [ConditionProviders.SCP:D] "^(.*?)nextUserAlarmTime(.*?)13:30:00" |
| Clock | "create alarm at 17:30 pm" | 11 | System log | [ConditionProviders.SCP:D] "^(.*?)nextUserAlarmTime(.*?)17:30:00" |
| Clock | "create alarm at 20:30 pm" | 11 | System log | [ConditionProviders.SCP:D] "^(.*?)nextUserAlarmTime(.*?)20:30:00" |
| Clock | "create alarm at 23:30 pm" | 11 | System log | [ConditionProviders.SCP:D] "^(.*?)nextUserAlarmTime(.*?)23:30:00" |
| Clock | "create alarm at 10:30 am on every weekday" | 11 | App data (Database) | [/data/user_de/0/com.google.android.deskclock/ databases/alarms.db] {"hour"=10,"minutes"=30, "daysofweek"=31} |
| Clock | "create alarm at 10:30 am on every midweek" | 12 | App data (Database) | [/data/user_de/0/com.google.android.deskclock/ databases/alarms.db] {"hour"=10,"minutes"=30, "daysofweek"=15} |
| Clock | "delete alarm at 9 am and create alarm at 10:30 am" | 12 | SApp data (Database) | [/data/user_de/0/com.google.android.deskclock/ databases/alarms.db] {"hour"=10,"minutes"=30, "daysofweek"=15} |
| Clock | "turn on alarm at 9 am in clock and increase alarm volume in setting" | 12 | App data (Database) + System setting | [/data/user_de/0/com.google.android.deskclock/ databases/alarms.db] {"hour"=10,"minutes"=30, "daysofweek"=15, "enabled"=1 } |
| Clock | "create alarm at 10:30 am on every weekend" | 14 | App data (Database) | [/data/user_de/0/com.google.android.deskclock/ databases/alarms.db] {"hour"=10,"minutes"=30, "daysofweek"=96} |
| Clock | "create alarm at 13:30 pm and another alarm 2 hours before it)" | 14 | App data (Database) | [/data/user_de/0/com.google.android.deskclock/ databases/alarms.db] [{"hour"=13,"minutes"=30}, {"hour"=11,"minutes"=30}] |
| Clock | "create alarm at 13:30 pm in clock and increase alarm volume in setting" | 14 | App data (Database) + System setting | [/data/user_de/0/com.google.android.deskclock/ databases/alarms.db] {"hour"=13,"minutes"=30} |
| Clock | "create alarm at 13:30 pm in clock and increase alarm volume in setting" | 15 | App data (Database) + System setting | [/data/user_de/0/com.google.android.deskclock/ databases/alarms.db] [{"hour"=13,"minutes"=30}, {"hour"=15,"minutes"=30}] |
| Clock | "create alarm at 13:30 pm on every weekend and increase alarm volume in setting" | 16 | App data (Database) + System setting | [/data/user_de/0/com.google.android.deskclock/ databases/alarms.db] {"hour"=13,"minutes"=30, "daysofweek"=31} |
| Clock | "create alarm at 13:30 pm on every weekend" | 16 | App data (Database) | [/data/user_de/0/com.google.android.deskclock/ databases/alarms.db] {"hour"=10,"minutes"=30, "daysofweek"=96} |

Table 8: Comprehensive list of tasks.

| Application | Task instruction | Step limit | Success criteria (source) | Success criteria (detail) |
|---|---|---|---|---|
| Clock | "create alarm at 10:30 am in clock and increase alarm volume in setting" | 18 | System setting | [/data/user_de/0/com.google.android.deskclock/ databases/alarms.db] {"hour"=10,"minutes"=30} |
| Clock | "create alarm at 13:30 pm and another alarm 2 hours after it" | 18 | System setting | [/data/user_de/0/com.google.android.deskclock/ databases/alarms.db] {"hour"=10,"minutes"=30} |
| Contacts | "open the contact app" | 4 | System log | [ActivityManager:I] "^(.*?)Start proc(.*?)com.android.contacts" |
| Contacts | "activate the insert page in contact" | 5 | System log | [ActivityManager:I] "^(.*?)START(.*?)INSERT(.*?) ContactEditorActivity" |
| Contacts | "activate the edit page in contact" | 5 | System log | [ActivityManager:I] "^(.*?)START(.*?)EDIT(.*?) ContactEditorActivity" |
| Files | "open the file manager app" | 4 | System log | [ActivityTaskManager:I] ^(.*?)START(.*?)files.FilesActivity" |
| Files | "list audio files in file manager" | 5 | System log | [DirectoryFragment:D] "^(.*?)Showing directory(.*?)audio(.*?)root" |
| Files | "list image files in file manager" | 5 | System log | [DirectoryFragment:D] "^(.*?)Showing directory(.*?)images" |
| Files | "list video files in file manager" | 5 | System log | [DirectoryFragment:D] "^(.*?)Showing directory(.*?)videos" |
| Files | "list download files in file manager" | 5 | System log | [DirectoryFragment:D] "^(.*?)Showing directory(.*?)download" |
| Gmail | "open Gmail" | 4 | System log | [ActivityTaskManager:I] "^(.*?)START(.*?)com.google.android.gm" |
| Instagram | "open Instagram" | 4 | UI elements | {"id": "com.instagram.android:id/feed_tab", "selected": "true"} |
| Instagram | "go to my profile in Instagram" | 5 | UI elements | {"id": "com.instagram.android:id/profile_tab", "selected": "true"} |
| Instagram | "go to reels tab in Instagram" | 5 | UI elements | {"id": "com.instagram.android:id/clips_tab", "selected": "true"} |
| Instagram | "go to search tab in Instagram" | 5 | UI elements | {"id": "com.instagram.android:id/search_tab", "selected": "true"} |
| Maps | "open the map app" | 4 | System log | [ActivityTaskManager:I] "^(.*?)START (.*?)com.google.android.maps.MapsActivity" |
| Messages | "open the message app" | 4 | System log | [ActivityTaskManager:I] "^(.*?)START (.*?)com.google.android.apps.messaging" |
| Messages | "start chatting in message" | 5 | System log | [BugleUsageStatistics:I] "^(.*?)BUGLE CREATE(.*?)DEFAULT" |
| Phone | "open the phone app" | 4 | System log | [Dialer:I] "^(.*?)MainActivity.onCreate" |
| Phone | "call 911" | 9 | System log | [Telecom:I] "^(.*?)Emergency number detected" |
| Phone | "call 11489" | 11 | UI elements | [{"id":"com.android.dialer:id/incall_end_cal", "enabled": "true"}, {"id":"com.android. dialer:id/contactgrid_contact_name", "text": "11489"}] |
| Phone | "call 311311" | 12 | UI elements | [{"id":"com.android.dialer:id/incall_end_cal", "enabled": "true"}, {"id":"com.android. dialer:id/contactgrid_contact_name", "text": "311311"}] |

Table 9: Comprehensive list of tasks.

| Application | Task instruction | Step limit | Success criteria (source) | Success criteria (detail) |
|---|---|---|---|---|
| Phone | "call 123-4578" | 13 | UI elements | [{"id":"com.android.dialer:id/incall_end_cal", "enabled": "true"}, {"id":"com.android. dialer:id/contactgrid_contact_name", "text": "1234578"}] |
| Phone | "call 223-4458" | 13 | UI elements | [{"id":"com.android.dialer:id/incall_end_cal", "enabled": "true"}, {"id":"com.android. dialer:id/contactgrid_contact_name", "text": "223-4458"}] |
| Phone | "call 402-7717" | 13 | UI elements | [{"id":"com.android.dialer:id/incall_end_cal", "enabled": "true"}, {"id":"com.android. dialer:id/contactgrid_contact_name", "text": "402-7717"}] |
| Phone | "call 766-3394" | 13 | UI elements | [{"id":"com.android.dialer:id/incall_end_cal", "enabled": "true"}, {"id":"com.android. dialer:id/contactgrid_contact_name", "text": "766-3394"}] |
| Phone | "call 987-6654" | 13 | UI elements | [{"id":"com.android.dialer:id/incall_end_cal", "enabled": "true"}, {"id":"com.android. dialer:id/contactgrid_contact_name", "text": "987-6654"}] |
| Phone | "call the national weather service (301-713-0622)" | 14 | UI elements | [{"id":"com.android.dialer:id/incall_end_cal", "enabled": "true"}, {"id":"com.android. dialer:id/contactgrid_contact_name", "text": "(301)713-0622"}] |
| Phone | "call the social security administration (800-772-1213)" | 14 | UI elements | [{"id":"com.android.dialer:id/incall_end_cal", "enabled": "true"}, {"id":"com.android. dialer:id/contactgrid_contact_name", "text": "(800)772-1213"}] |
| Phone | "call 26-445-1193" | 15 | UI elements | [{"id":"com.android.dialer:id/incall_end_cal", "enabled": "true"}, {"id":"com.android. dialer:id/contactgrid_contact_name", "text": "(264)451-193"}] |
| Phone | "call the US national contact center (800-333-4636)" | 16 | UI elements | [{"id":"com.android.dialer:id/incall_end_cal", "enabled": "true"}, {"id":"com.android. dialer:id/contactgrid_contact_name", "text": "(800)333-4636"}] |
| Phone | "call the white house (202-456-1111)" | 17 | UI elements | [{"id":"com.android.dialer:id/incall_end_cal", "enabled": "true"}, {"id":"com.android. dialer:id/contactgrid_contact_name", "text": "(202)456-1111"}] |
| Photos | "open the photos app" | 4 | System log | [ActivityTaskManager:I] ^(.*?)START(.*?)com.google.android.apps.photos |
| Settings | "open the setting app" | 4 | System log | [ActivityManager:I] "^(*.?)Start proc(.*?)com.android.settings.Settings" |
| Settings | "turn on airplane mode" | 5 | System log | [PhoneGlobals:I] "^(.*?)Turning radio off(.*?)airplane" |
| Settings | "turn off airplane mode" | 5 | System log | [PhoneGlobals:I] "^(.*?)Turning radio on(.*?)airplane" |
| Settings | "turn on wifi" | 5 | System log | [WifiService:I] "^(.*?)setWifiEnabled (.*?)com.android.settings(.*?)enable=true" |
| Settings | "turn off wifi" | 5 | System log | [WifiService:I] "^(.*?)setWifiEnabled (.*?)com.android.settings(.*?)enable=false" |

Table 10: Comprehensive list of tasks.

| Application | Task instruction | Step limit | Success criteria (source) | Success criteria (detail) |
|---|---|---|---|---|
| Settings | "decrease screen brightness in setting" | 6 | System log | [DisplayPowerController:V] "^(.*?)Brightness(.*?)changing(.*?)manual" |
| Settings | "go to app info list in setting" | 6 | System log | [SettingsActivity:D] "^(.*?)Switching (.*?)android.settings(.*?)System log" |
| Settings | "go to bluetooth setting" | 6 | System log | [PrefCtrlListHelper:D] "^(.*?)android.settings.bluetooth.BluetoothDevice" |
| Settings | "toggle dark theme in setting" | 6 | System log | [SettingsProvider:V] "^(.*?)content(.*?)settings(.*?)dark(.*?)mode" |
| Settings | "toggle vibrate for calls in setting" | 6 | System log | [SettingsProvider:V] "^(.*?)vibrate(.*?)when(.*?)ringing" |
| Settings | "increase media volume in setting" | 6 | System log | [SettingsProvider:V] "^(.*?)MEDIA" |
| Settings | "increase call volume in setting" | 6 | System log | [SettingsProvider:V] "^(.*?)CALL" |
| Settings | "increase ring volume in setting" | 6 | System log | [SettingsProvider:V] "^(.*?)MUSIC" |
| Settings | "increase alarm volume in setting" | 6 | System log | [SettingsProvider:V] "^(.*?)ALARM" |
| Settings | "go to 'add a language' page in setting" | 7 | System log | [ActivityTaskManager:I] "^(.*?)LocalePicker" |
| Snapseed | "open Snapseed" | 4 | UI elements | {"id": "com.niksoftware.snapseed:id/logo_view", "enabled": "true"} |
| Snapseed | "open image in Snapseed" | 6 | UI elements | {"id": "com.niksoftware.snapseed:id/looks_button", "selected": "true"} |
| Snapseed | "set dark theme in Snapseed" | 7 | App data (xml) | [/data/data/com.niksoftware.snapseed/ shared_prefs/Preferences.xml] {"pref_appearance_use_dark_theme"="true"} |
| Snapseed | "open image and apply portrait filter in Snapseed" | 7 | UI elements | {"-android uiautomator": " new UiSelector().text("Portrait")","selected": "true"} |
| Snapseed | "set format quality to JPG 100% in Snapseed" | 9 | App data (xml) | [/data/data/com.niksoftware.snapseed/ shared_prefs/Preferences.xml] {"pref_export_setting_compression"="100"} |
| Snapseed | "set image sizing to 2000 px" | 9 | App data (xml) | [/data/data/com.niksoftware.snapseed/ shared_prefs/Preferences.xml] {"pref_export_setting_long_edge"="2000"} |
| Snapseed | "open image and go to tools tab in Snapseed" | 9 | UI elements | {"id": "com.niksoftware.snapseed:id/tools_button", "selected": "true"} |
| Snapseed | "open image and apply noir S03 filter in Snapseeed" | 10 | UI elements | {"-android uiautomator": "new UiSelector().text("S03")", "selected": "true"} |
| Snapseed | "apply noire S03 filter to an image after setting dark theme in Snapseed" | 13 | App data (xml) + UI elements | {"-android uiautomator": "new UiSelector().text("S03")", "selected": "true"}, [/data/data/com.niksoftware.snapseed/ shared_prefs/Preferences.xml] "pref_appearance_use_dark_theme"="true" |
| Snapseed | "apply noire S03 filter to an image after setting format quality to JPG 100% in Snapseed" | 14 | App data (xml) + UI elements | {"-android uiautomator": "new UiSelector().text("S03")", "selected": "true"}, [/data/data/com.niksoftware.snapseed/ shared_prefs/Preferences.xml] "pref_export_setting_compression"="100" |
| Snapseed | "apply noire S03 filter to an image after setting image sizing to 2000 px in Snapseed" | 14 | App data (xml) + UI elements | {"-android uiautomator": "new UiSelector().text("S03")", "selected": "true"}, [/data/data/com.niksoftware.snapseed/ shared_prefs/Preferences.xml] "pref_export_setting_long_edge"="2000" |
| Walmart | "open Walmart" | 4 | UI elements | {"id": "com.walmart.android:id/navigation _shop", "enabled": "true"} |
| Walmart | "go to account tab in Walmart" | 5 | UI elements | {"id": "com.walmart.android:id/navigation _account", "selected": "true"} |

Table 11: Comprehensive list of tasks.

| Application | Task instruction | Step limit | Success criteria (source) | Success criteria (detail) |
|---|---|---|---|---|
| Walmart | "go to my cart in Walmart" | 5 | UI elements | {"id": "com.walmart.android:id/cart_fragment _constraint_layout", "selected": "true"} |
| Walmart | "go to my items tab in Walmart" | 5 | UI elements | {"id": "com.walmart.android:id/navigation _my_items", "selected": "true"} |
| Walmart | "go to search tab in Walmart" | 5 | UI elements | {"id": "com.walmart.android:id/navigation _search", "selected": "true"} |
| Walmart | "go to services tab in Walmart" | 5 | UI elements | {"id": "com.walmart.android:id/navigation _services", "selected": "true"} |
| Walmart | "go to grocery category and show subcategories in Walmart" | 7 | UI elements | {"id": "com.walmart.android:id/category_ container_title", "text": "Grocery"} |
| Walmart | "go to store map in Walmart" | 7 | UI elements | {"id": "com.walmart.android:id/instoremaps_ webview_container", "displayed": "true"} |
| Wikipedia | "open Wikipedia" | 4 | UI elements | {"id": "org.wikipedia:id/nav_tab_explore", "selected": "true"} |
| Wikipedia | "go to saved tab in Wikipedia" | 5 | UI elements | {"id": "org.wikipedia:id/nav_tab_reading_lists", "selected": "true"} |
| Wikipedia | "go to search tab in Wikipedia" | 5 | UI elements | {"id": "org.wikipedia:id/nav_tab_search", "selected": "true"} |
| Wikipedia | "disable the top 1 and 'randomizer' topics in the feed customization settings on Wikipedia and go back to the feed" | 10 | App data (xml) + UI elements | [/data/data/org.wikipedia/shared_prefs/org. wikipedia_preferences.xml] {"feedCardsEnabled": "[false,true,true,true,true,true,false,true,true,true]"}, {"id": "org.wikipedia:id/nav_tab_explore", "selected": "true"} |
| Wikipedia | "disable the top 2 topics in the feed customization settings on Wikipedia and go back to the feed" | 10 | App data (xml) + UI elements | [/data/data/org.wikipedia/shared_prefs/org. wikipedia_preferences.xml] {"feedCardsEnabled": "[false,false,true,true,true,true,true,true,true,true]"}, {"id": "org.wikipedia:id/nav_tab_explore", "selected": "true"} |
| Wikipedia | "disable the top 2 and 'randomizer' topics in the feed customization settings on Wikipedia and go back to the feed" | 11 | App data (xml) + UI elements | [/data/data/org.wikipedia/shared_prefs/org. wikipedia_preferences.xml] {"feedCardsEnabled": "[false,false,true,true,true,true,false,true,true,true]"}, {"id": "org.wikipedia:id/nav_tab_explore", "selected": "true"} |
| Wikipedia | "decrease the text size to 50% in Wikipedia" | 11 | App data (xml) | [/data/data/org.wikipedia/shared_prefs/org. wikipedia_preferences.xml] {"textSizeMultiplier": -5} |
| Wikipedia | "disable the 'show link previews', 'top read' feed settings, and return to the feed on Wikipedia" | 11 | App data (xml) + UI elements | [/data/data/org.wikipedia/shared_prefs/org. wikipedia_preferences.xml] {"feedCardsEnabled": [false,true,true,true,true,true,true,true,true,true]} {"id": "org.wikipedia:id/nav_tab_explore", "selected": "true"} |
| Wikipedia | "disable the topics that are related to 'history' in the feed customization settings on Wikipedia and go back to the feed" | 12 | App data (xml) + UI elements | [/data/data/org.wikipedia/shared_prefs/org. wikipedia_preferences.xml] {"feedCardsEnabled": [true,true,true,false,true,false,true,true,true,true]} {"id": "org.wikipedia:id/nav_tab_explore", "selected": "true"} |
| Wikipedia | "disable the topics that include 'day' in their names in the feed customization settings on Wikipedia and go back to the feed" | 13 | App data (xml) + UI elements | [/data/data/org.wikipedia/shared_prefs/org. wikipedia_preferences.xml] {"feedCardsEnabled": [true,true,false,true,true,false,true,true,true,true]} {"id": "org.wikipedia:id/nav_tab_explore", "selected": "true"} |

Table 12: Comprehensive list of 131 tasks.

| Application | Task instruction | Step limit | Success criteria (source) | Success criteria (detail) |
|---|---|---|---|---|
| Wikipedia | "disable the topics with odd-numbered indices in the feed customization settings on Wikipedia and go back to the feed" | 13 | App data (xml) + UI elements | [/data/data/org.wikipedia/shared_prefs/org.wikipedia_preferences.xml] {"feedCardsEnabled": "[false,true,false,true,false,true,false,true,true,true]"}, {"id": "org.wikipedia:id/nav_tab_explore", "selected": "true"} |
| Wikipedia | "disable the topics with even-numbered indices in the feed customization settings on Wikipedia and go back to the feed" | 14 | App data (xml) + UI elements | [/data/data/org.wikipedia/shared_prefs/org.wikipedia_preferences.xml] {"feedCardsEnabled": [true,false,true,false,true,false,true,false,true,true]} {"id": "org.wikipedia:id/nav_tab_explore", "selected": "true"} |
| Wikipedia | "disable the topics with prime-numbered indices in the feed customization settings on Wikipedia and go back to the feed" | 14 | App data (xml) + UI elements | [/data/data/org.wikipedia/shared_prefs/org.wikipedia_preferences.xml] {"feedCardsEnabled": [true,false,false,true,false,true,false,true,true,true]} {"id": "org.wikipedia:id/nav_tab_explore", "selected": "true"} |
| Wikipedia | "increase the text size to 180% in Wikipedia" | 15 | App data (xml) | [/data/data/org.wikipedia/shared_prefs/org.wikipedia_preferences.xml] {"textSizeMultiplier": -5} |
| Wikipedia | "disable featured article feed, decrease the text size to 50%, and return to the feed on Wikipedia" | 17 | App data (xml) + UI elements | [/data/data/org.wikipedia/shared_prefs/org.wikipedia_preferences.xml] {"textSizeMultiplier": -5, "feedCardsEnabled": [false,true,true,true,true,true,true,true,true,true]} {"id": "org.wikipedia:id/nav_tab_explore", "selected": "true"} |
| Wikipedia | "disable featured article feed, increase the text size to 180%, and return to the feed on Wikipedia" | 19 | App data (xml) + UI elements | [/data/data/org.wikipedia/shared_prefs/org.wikipedia_preferences.xml] {"textSizeMultiplier": 8, "feedCardsEnabled": [false,true,true,true,true,true,true,true,true,true]} {"id": "org.wikipedia:id/nav_tab_explore", "selected": "true"} |
| Youtube | "open Youtube" | 4 | System log | [ActivityTaskManager:I] "^(.*?)START(.*?)com.google.android.youtube" |

Table 13: Comprehensive list of 131 tasks.

We categorize each task based on the application into five groups: System, Web/Shopping, Communication, Utility, and Event. The group System includes tasks using applications of Files and Settings. The group Web/Shopping includes tasks using applications of Chrome, Google, Walmart, and Wikipedia. The group Communication includes tasks using applications of Contacts, Gmail, Instagram, Message, Phone, and Youtube. The group Utility includes tasks using applications of Calculator, Camera, Maps, Photos, and Snapseed. The group Event includes tasks using applications of Calendar and Clock.

Some of the target applications in the daily tasks require a log-in process. We warn that exploiting private accounts on these tasks may cause a leak of personal information. In creating tasks that may incorporate log-in (such as third-party apps like Instagram), we set the tasks to be concise and maximum step limits to be as small as possible. We provide the list of tasks that demand log-in in Table 14. The "Account Association" column in Table 14 stands for the platform or website where the account should belong.

| Application | Task instruction | Account Association |
|---|---|---|
| Instagram | "open Instagram" | Instagram |
| Instagram | "go to my profile in Instagram" | Instagram |
| Instagram | "go to reels tab in Instagram" | Instagram |
| Instagram | "go to search tab in Instagram" | Instagram |

Table 14: The list of tasks requiring log-in process.

### B.2 EXEMPLARY DEMONSTRATIONS ON REPRESENTATIVE TASKS

In our experiments, we select six representative tasks. The tasks are selected to cover various functionalities, such as navigating pages (e.g., tab in the clock application or different setting pages in the setting application) and manipulating various UI elements (e.g., checkbox, dial pad, time pickers, etc.). On each task, we display the successful demonstration in Figure 9.

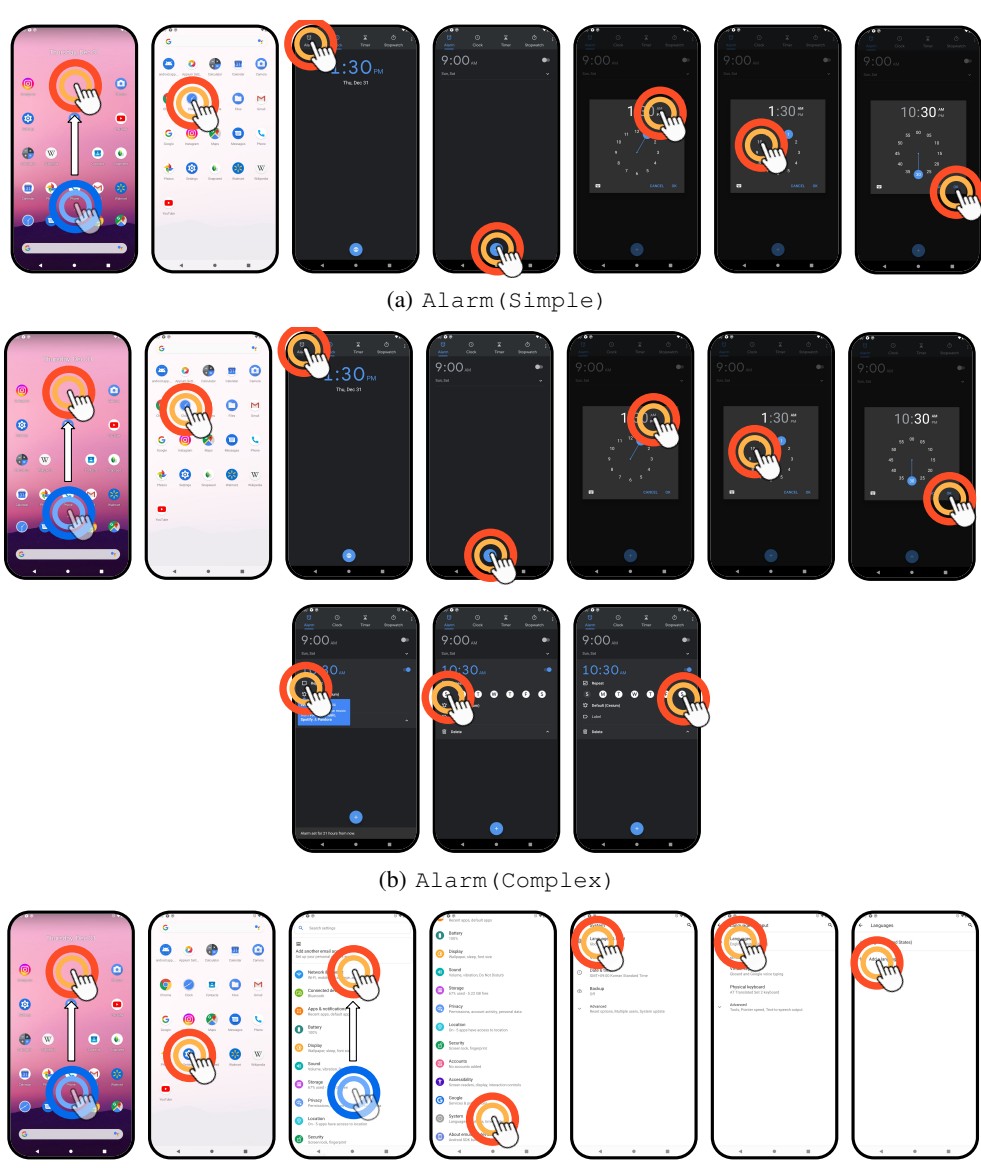

(a) Alarm(Simple)

(b) Alarm(Complex)

(c) Language

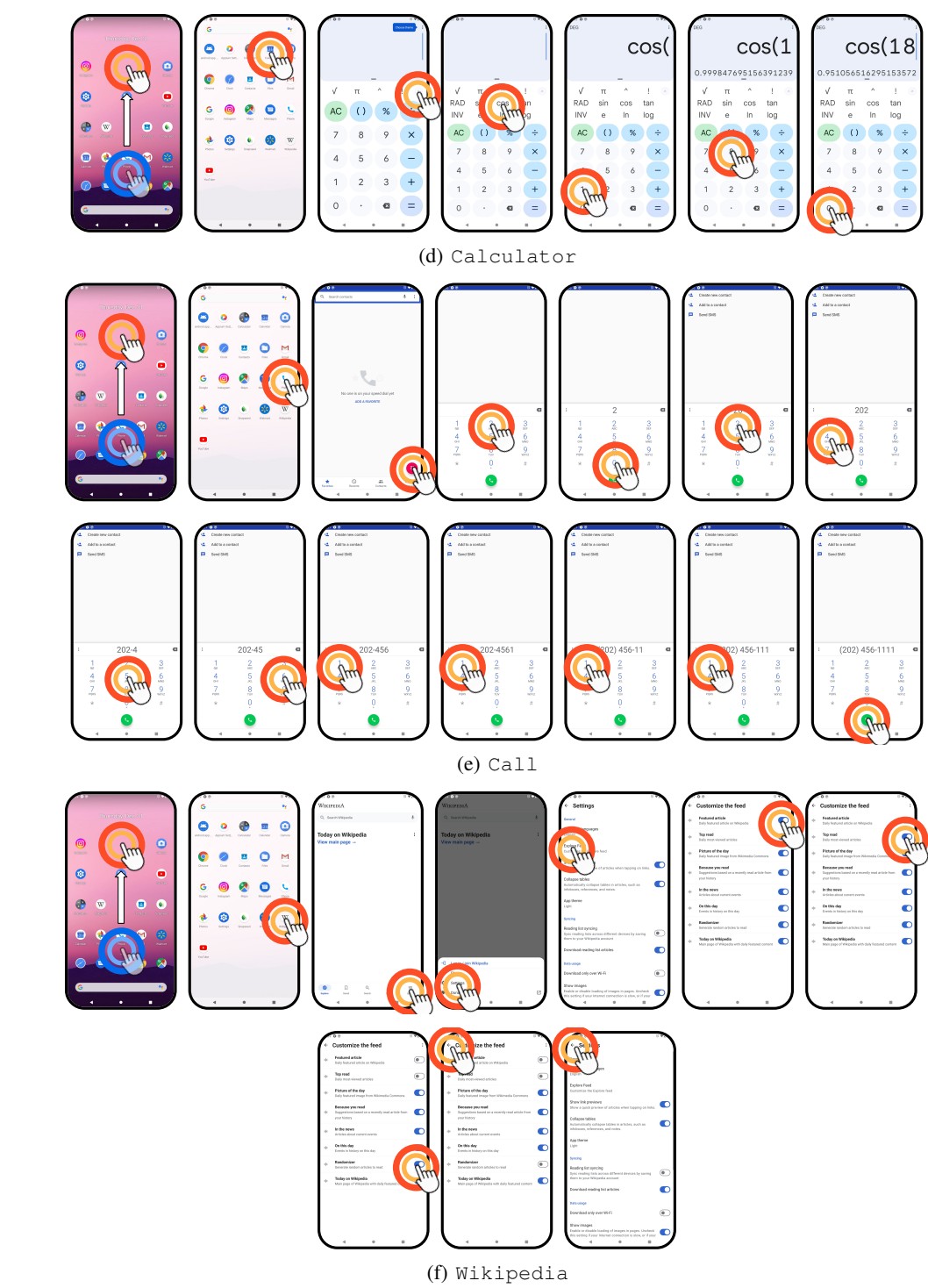

Figure 9: Examples of human expert demonstrations of six representative tasks. The blue and red cursors linked with a white arrow identify the swiping action, while the red cursor alone identifies the tapping action.

## C AGENT DETAILS

### C.1 OVERVIEW OF AGENTS EMPLOYING FOUNDATION MODELS

We provide an overall workflow (illustrated in Figure 10) of agents employing foundation models, especially with LLM. To employ LLM to develop the agents, we build a prompt based on task instruction and text-based observation. The text-based observation describes the screen layout information. We provide the prompt to the LLM and obtain the response. The response includes the text-based action chosen from a set of action options (which is provided in the prompt) by the LLM. The action converter extracts the legal action from the text-based action. The environment which is realized as an Android emulator, then, processes the provided action.

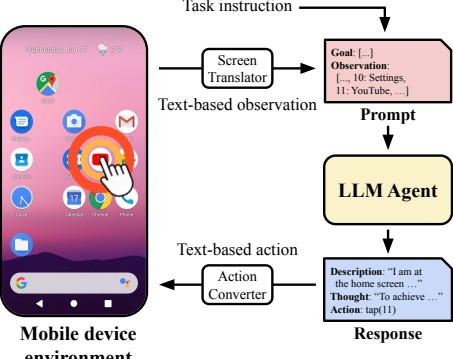

Figure 10: Illustration of the LLM agent controlling mobile devices. LLM agents interact with the environments through an additional screen translator and action converter, to obtain text-based observations and control with text-based actions.

### C.2 PROMPT DETAILS FOR AGENTS WITH FOUNDATION MODELS

For the agents employing LLMs or MLLMs, we use a complete prompt format described in Table 15. The role description informs the agents with general instructions about the problem, i.e., device control problem. The possible actions are provided as callable functions: options of tapping an element in the list, swiping the screen, and pressing the three navigation buttons over the screen with action press("BACK"), action press("HOME"), and action press("OVERVIEW"). The output format is designed to integrate the Chain-of-Thought (CoT) technique (Wei et al., 2022). In our experiments, we observe that the agents using GPT-4o, Gemini-1.5-pro, and Llama-3 can accurately generate a description of the screen layout, rationale on the current situation and next step, and actions in a function call following the format with our prompt. We also provide a list of previous actions taken by the agents.

For the text-based observations, we provide a numeric tag and detailed information for each UI element, including resource ID information, class information, content description information, text information, and additional information such as checked information or bounding box location information. The resource ID and class information capture the coarse-grained information on the UI element, such as the type of UI element. The content description and text information serve as fine-grained information on the UI element, such as the specific name of the application. The checked information and bounding box location information supply additional information on the UI element.

For few-shot learning of agents with foundation models, we include a pre-defined number of examples hinting correct actions to take. Specifically, to build a prompt with few-shot examples, the [few shot prompt] part in Table 15 is replaced with the text illustrating the human demonstration. Table 16 shows an exemplary few show prompts, with one trajectory of the human expert demonstration. In our experiments, we provide a whole trajectory performed by the human expert on solving each task as a few-shot example by sampling from the set of trajectories performed in various training environments. For Llama-3, due to the limited context length, we sample observation-action pairs from the trajectory.

> You are an agent that is trained to perform daily tasks on digital devices, such as smartphones.
>
> You are given a goal task instruction to accomplish, an observation from the environment, and previous actions you have taken (up to 4 past steps).
>
> The observation is a screen description parsed from the Android view hierarchy which contains the numeric tag and relevant information (e.g., text description) of each UI element.
>
> For the response, you need to think and call the function needed to achieve the goal task instruction.
> Your output should include three parts in the given format:
> - Description: <Describe what you observe in the input>
> - Thought: <Provide a rationale on the next action you should take to complete the task>
> - Action: <Select an action option in the format of a function call with the correct parameters. You cannot output anything else except a function call.>
>
> For the action, you need to select an action option by calling one of the following functions to control the digital device:
> 1. dual-gesture(touch y: float, touch x: float, lift y: float, lift x: float): This function is used to operate a dual-gesture action. A dual-gesture comprises four floating-point numeric values between 0 and 1, indicating a normalized location of the screen in each of the x-y coordinates. A dual-gesture action is interpreted as touching the screen at the location of (touch y, touch x) and lifting at the location of (lift y, lift x). The dual-gesture action indicates a tapping action if the touch and lift locations are identical, but a swiping action if they differ. A simple use case is dual-gesture(0.5, 0.5, 0.5, 0.5) to tap the center of the screen.
> 2. tap(numeric tag: int): This function is used to tap a UI element shown on the digital device screen. "numeric tag" is a tag assigned to a UI element shown on the digital device screen. A simple use case can be tap(5), which taps the UI element labeled with the number 5.
> 3. swipe(direction: str): This function is used to swipe on the digital device screen. "direction" is a string that represents one of the four directions: up, down, left, right. "direction" must be wrapped in double quotation marks. A simple use case is swipe("up"), which can be used to open the app list on the home screen.
> 4. press("HOME"): This function is used to press the home button.
> 5. press("BACK"): This function is used to press the back button.
> 6. press("OVERVIEW"): This function is used to press the overview button.
> You can only take one action at a time, so please directly call the function.
> Please never take action besides the options provided.
>
> Goal: [task instruction].
>
> {few shot prompt}
>
> Previous actions: [previous actions]
> Current observation: [current observation]
> Answer:

Table 15: Prompts used for LLM agents and MLLM agents. Parts for [...] are filled for different environments (including the tasks). Parts for {...} are filled in according to different experiments, as few-shot examples are optional.

> Below illustrates the example of human experts.
> Each example is a full trajectory from the beginning to the end of the task completion.
> Each observation from the environment and corresponding action taken by the expert is described as:
> - Observation: <An observation from the environment>
> - Action: <An action taken by the human expert>
>
> Demonstration Example:
> - Observation: [{'numeric_tag': 0, 'resource-id': '', 'class': 'View', 'description': 'Appslist', ..., 'checked': False}, [...], {'numeric_tag': 38, 'resource-id': '', 'class': 'FrameLayout', 'description': '', ..., 'checked':False}]
> - Action: swipe("up")
> {...}
> - Observation: [{'numeric_tag': 0, 'resource-id': '', 'class': 'FrameLayout' 'description': '', ..., 'checked': False}, [...], {'numeric_tag': 83, 'resource-id': '', 'class': 'LinearLayout', 'description': '', ..., 'checked': False}]
> - Action: tap(74)

Table 16: An exemplary few-shot prompt with one trajectory of human expert demonstration. For the attributes of UI elements in the observation, the value is set to be '' if it is unavailable (e.g., 'resource-id' of the first element in the first observation). Parts for [...] are filled with a list of descriptions for UI elements. Parts for {...} are filled with a list of intermediate steps in the expert demonstration.

## C.3 ARCHITECTURE DESIGN FOR CUSTOM AGENTS TRAINED WITH VLM ENCODER

The network architecture for custom agents is composed of three components: encoder, attention module, and action head. Given the task instruction $c$ and the screen image $o_t \in \mathbb{R}^{3 \times 256 \times 128}$ at each timestep $t$, custom agents generate a screen-touching action $a_t \in \mathbb{R}^{385}$. Among the screen-touching action $a_t$, the first values correspond to tapping pre-defined locations (14×27 bins) on the screen, four values to swiping directions (up, down, right, left), and the last three values to pressing buttons (back, home, overview).

Custom agents use visual and text encoders to represent screen images $o_t$ and task instruction $c$, respectively. The visual encoder embeds visual feature $e_{o_t} \in \mathbb{R}^d$ from the observation $o_t$, and the text encoder extracts features $e_c \in \mathbb{R}^d$ from the task instruction $c$. For the visual encoder, we use EfficientNet-b0 (Tan & Le, 2019) pre-trained with ImageNet followed by an adaption layer using a fully connected layer to adapt the output channel to hidden dimension $d$ (Liu et al., 2023). For the text encoder, we use a pre-trained text encoder of Text-to-Text Transfer Transformer (Zhan & Zhang, 2023) which is trained with Android-in-the-wild (Rawles et al., 2023), which is a dataset composed of demonstrations for solving Android device control problems. The text encoder is kept frozen during the training process. The hidden dimension $d$ is set to value of 768 for both visual embedding and text embedding.

The attention module, then, fuses the visual feature $e_{o_t}$ and text feature $e_c$ into a single vision-language embedding $e_{\text{fused}} \in \mathbb{R}^d$. Especially, we use multi-head attention layer (Vaswani et al., 2017) for cross-attention, with $e_c$ given as query and $e_{o_t}$ given as key and value. Given the fused feature $e_{\text{fused}}$, the action heads predict the action $a_t$, by applying the three fully connected (FC) layers with the size of (1024, 1024, 385), respectively.

# D EXPERIMENT DETAILS

## D.1 DEMONSTRATIONS COLLECTION

The demonstrations are collected by human experts (authors). We allow the demonstrators to be accustomed to the environment interfaces by letting to interact with the environments (i.e., Android emulators). In this period demonstrators as asked to manipulate the emulators without specifying any certain task or to solve several random daily tasks. Then, we instruct the demonstrators to perform demonstrations on the target representative tasks. We ensure them to operate the emulator optimally and consistently along different device configurations as much as possible. The instruction for the target tasks is the same as the task instruction provided to the agents. For few-shot examples of agents employing foundation models, we ask demonstrators to use discrete actions, except when the task is impossible to solve without using continuous actions (i.e., dual-gesture actions).

For the few-shot learning of LLM and MLLM agents and training of custom agents, we collect human expert demonstrations. The collectors (graduate students) are instructed to complete the six representative tasks in each *training* environment. The definitions of action space for the collected demonstration are in two modes: the action space defined with action options (for agents using foundation models) and the action space as a set of dual-gesture actions (for custom agents). The end of each episode is determined by the success detector we implement. We exclude the demonstrations with failures. In our experiments, all the exemplary actions for agents employing foundation models are in discrete actions.

For the experiments in Section 4.2, we exploit training environments with ID numbers from 000 to 034. Hence, a total number of 210 trajectories of demonstrations are prepared. For agents using closed-source foundation models (i.e., GPT-4o and Gemini-1.5-pro), a whole trajectory by the human expert on solving a target task is sampled as a few-shot example. For agents employing open-source LLM (i.e., Llama-3), a pair of observation and action is provided randomly sampled from the exemplary trajectories as few-shot examples, similar to several prior methods (Zhang et al., 2023; Rawles et al., 2023), owing to the limited context length. We exploit three few-shot examples for each step. For custom agents, each triplet (task instruction, observation, action) in the trajectories is used as a data point for composing the training batch. For the experiments on the effect of data diversity on custom agents (Section 4.3), we leverage varying numbers of training environments 7

and 35 where the corresponding ID numbers of the environments are from 000 to 006 and from 000 to 034, respectively. The total number of demonstrations for each setting is 42 and 210, respectively.

## D.2 CONFIGURATION DETAILS FOR AGENTS WITH FOUNDATION MODELS

For the experiments on agents with foundation models, we set the configurations for the foundation models. We set the temperature to 0.0 and top-p with the default value of 1.0 (as altering only either temperature or top-p from the default setting is suggested) for GPT-4o. We set the temperature to 0.0 and top-p to 1.0 for Gemini-1.5-Pro. We set the temperature to 0.1, top-p to 1.0, top-k to 42, num-return-sequences to 1, and no-repeat-ngram-size to 4 for Llama-3. The maximum output tokens for all models are set to 256. The values of the parameters unspecified are set to default. We experiment with three different runs. We note that we fix seeds for consistent output generation across the runs when employing the closed-source LLMs, while they do not provide deterministic generation by default.

## D.3 TRAINING DETAILS FOR CUSTOM AGENTS

**Custom agents using fine-tuned Llama-3-8B** For the experiments on custom agents using fine-tuned Llama-3, we fine-tune the pre-trained `Llama-3-8B-Instruct` model over 15 epochs, using a batch size of 8 with gradients accumulation over 8 steps, sampled from a collection of 210 human demonstrations. We use the AdamW optimizer (Loshchilov, 2019) with a learning rate of 1e-6 with a cosine annealing scheduler (setting a warmup ratio of 0.01 and a weight decay of 1.0). In particular, we adopt the LoRA (Hu et al., 2022) technique, setting a rank value to 8 and the alpha to 16. Each training is conducted on 8 NVIDIA RTX 3090 GPUs and takes approximately three hours.

**Custom agents using VLM encoder** For the experiments on custom agents using VLM encoder, we train each multi-task policy over 4K steps with a batch size of 32, sampled from a collection of 210 human demonstrations. We use the Adam optimizer (Kingma & Ba, 2017) with a learning rate of 3e-4 and adopt a cosine annealing learning rate schedulers Each training is conducted on a single NVIDIA RTX A6000 GPU and takes approximately one hour.

## D.4 PERFORMANCES OF LLM AGENTS USING LLAMA-3-70B ON ALL TASKS

We evaluate LLM agents using Llama-3 (`meta-llama/Meta-Llama-3-70B-Instruct`), without fine-tuning, on all tasks in three representative test environments. Similar to the results with six representative challenging tasks, as shown in Figure 6, LLM agents using GPT-4o outperform the agents using Llama-3 on the evaluation with all tasks.

| Test Env ID | LLM Agent (GPT-4o) | LLM Agent (Gemini-1.5-pro) | LLM Agent (Llama-3) |
|---|---|---|---|
| 100 | $55.73 \pm 0.95$ | $39.69 \pm 0.36$ | $36.90 \pm 1.66$ |
| 101 | $48.85 \pm 0.95$ | $31.30 \pm 0.72$ | $19.34 \pm 0.55$ |
| 105 | $42.75 \pm 0.72$ | $25.45 \pm 1.81$ | $16.79 \pm 0.62$ |

Table 17: Success rates of LLM agents using Llama-3 on all tasks with three representative test environments. For easy comparison, we also show the performances of LLM agents using closed-source models (i.e., GPT-4o and Gemini-1.5-pro). LLM agents employing Llama-3 report the lowest performances in all test environments.

## D.5 PERFORMANCES OF CUSTOM AGENTS USING VLM ENCODER IN TRAINING ENVIRONMENTS

Figure 11 displays the differences in the success rates of custom agents in training and test environments. The challenges with diverse device configurations degenerate the performances of the custom agents. For example, on the `Language` task, the success rates decrease from higher than 90% in the training environments to less than 60% in the test environments. The differences between success rates in the training and test environments demonstrate the headroom for the generalization ability.

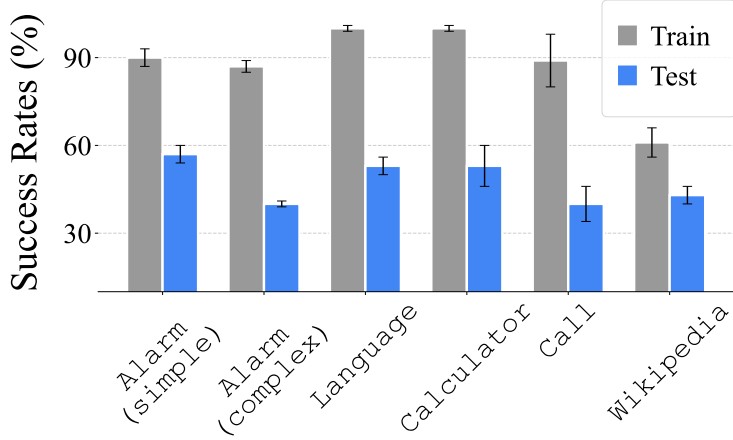

Figure 11: Success rates of custom agents trained with BC on training and test environments. The differences between the success rates demonstrate the headroom for the generalization ability.

### D.6 PERFORMANCES OF MLLM AGENTS WITH AND WITHOUT FEW-SHOT EXAMPLES

Table 18 shows the performances of MLLM agents using GPT-4o with and without few-shot examples (with the performances of LLM agents using GPT-4o for easy comparison). We observe a similar trend between LLM agents and MLLM agents. To be specific, the few-shot examples help MLLM agents on several tasks (e.g., Alarm(complex)), while they are not beneficial in some cases and even decrease the performances (e.g., Alarm(Simple)).

| | LLM Agent (zero-shot) | LLM Agent (few-shot) | MLLM Agent (zero-shot) | MLLM Agent (few-shot) |
|---|---|---|---|---|
| Alarm(simple) | $87 \pm 09$ | $83 \pm 07$ | $80 \pm 06$ | $63 \pm 09$ |
| Alarm(complex) | $33 \pm 07$ | $67 \pm 03$ | $23 \pm 03$ | $47 \pm 12$ |
| Language | $77 \pm 03$ | $80 \pm 06$ | $53 \pm 03$ | $77 \pm 03$ |
| Calculator | $17 \pm 03$ | $17 \pm 09$ | $13 \pm 03$ | $23 \pm 13$ |
| Call | $00 \pm 00$ | $30 \pm 10$ | $00 \pm 00$ | $07 \pm 07$ |
| Wikipedia | $30 \pm 06$ | $20 \pm 00$ | $27 \pm 03$ | $20 \pm 06$ |
| Average | $41 \pm 04$ | $49 \pm 05$ | $33 \pm 01$ | $39 \pm 03$ |

Table 18: Success rates of MLLM agents and LLM agents using GPT-4o with and without few-shot examples.

## E AGENTS TRAINED WITH REINFORCEMENT LEARNING

In this section, we explain further analysis with the agents employing policies trained with reinforcement learning. We, first, train the policies using the success signal as a reward. We, then, further study the efficacy of reward shaping for improving efficiency.

### E.1 EXPERIMENTAL SETUP

**Algorithm** To train custom agents trained with RL, we use double deep Q-network (DDQN) (Van Hasselt et al., 2016) and proximal policy optimization (PPO) Schulman et al. (2017). DDQN trains the two optimal Q functions $Q_1$ and $Q_2$ by minimizing the TD loss for the agent data sampled from the replay buffer $D$. As an example, we write the objective for $Q_1$(and its parameterization $Q_{\theta_1}$) as follows:

$$L_{\text{TD}}(Q_1) = \mathbb{E}_{(s,a,r,s') \sim D}[(r + \gamma Q_{\bar{\theta}_1}(s', a^*) - Q_{\theta_1}(s, a)^2].$$

where $a^* = \arg\max_{a'} Q_{\bar{\theta}_2}(s, a)$ and $\bar{\theta}$ denotes the target network for $\theta$, which in practice is replaced by the moving average of $\theta$. Especially, when calculating the TD target, we swap the maximizing

actions of the next state action values between $Q_1$ and $Q_2$ to prevent the overestimation of the values as presented in (Hasselt, 2010). To balance the exploration and the exploitation during the training, we employ the $\epsilon$-greedy technique to sample actions for collecting the training data. For the critic network architecture, we adopt the same visual encoder and text encoder backbone as the custom agent (using VLM encoder) and extract the fused feature $e_{\text{fused}}$. Finally, we add three FC layers comprising the dimensions of (1024, 1024, 385) to output the Q values for each action.

PPO (Schulman et al., 2017) builds upon the policy gradient method (Sutton & Barto, 2018) and uses a clipped surrogate objective. To be detailed, the objective is given by

$$L_{\text{PPO}}(\theta) = E_{\pi_{\text{old}}} \left[ \min \left( \text{clip}(r(\theta), 1 - \epsilon, 1 + \epsilon) A_t, r(\theta) A_t \right) \right],$$

where $A_t$ denotes the estimator for the future expected return and $r(\theta) = \pi_\theta(a|s)/\pi_{\text{old}}(a|s)$ corresponds to an importance sampling ratio between the current and the previous policy. Following (Schulman et al., 2017), we use generalized advantage estimator (GAE) (Schulman et al., 2016) in place of $A_t$. We use the same network architecture as the custom agent (using VLM encoder) for implementing $\pi_\theta$. For the value network $V$, we apply three FC layers that follow the size of (1024, 1024, 1) using the feature $e_{\text{fused}}$ produced by the pre-trained backbone.

**Task and environment**  We experiment RL with a sparse success signal using a task with the instruction "open Gmail" (denoted as `Gmail`) and `Language` task. We train at 10 training environments, whose IDs are 000, 001, 002, 003, 004, 005, 007, 008, 021, 022 and then evaluate under 10 test environments. We also experiment with dense reward setup using the task with the instruction "call 911" (denoted as `Call 911`) task under a training environment with id 000.

**Training procedure**  For DDQN, we update the critic network by sampling 1/4 transitions from the replay buffer that stores the successful history and the last from the failure buffer in every 5 episodes. For PPO, we apply one update using the on-policy samples for every 5 episodes and use the Adam optimizer with a learning rate of 2e-4. We use the hyperparameters of $\epsilon = 0.1$, $\gamma = 0.9$, and $\lambda = 0.9$ for both actor and critic unless otherwise specified. Each training is conducted on a single NVIDIA RTX A6000 GPU and Intel(R) Xeon(R) Gold 6426Y CPU. We note that the training procedure can be accelerated by vectorizing the training environments.

## E.2 RESULTS WITH RL TRAINING

We first train custom agents using RL on a `Gmail` task. We show the success rate in 10 training and test environments with three different seeds in Figure 12.

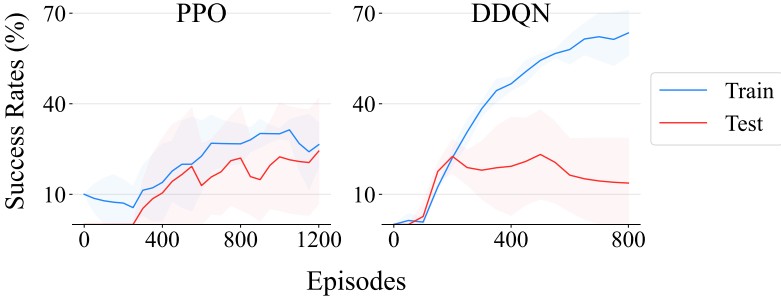

Figure 12: Training and test curves of DDQN and PPO algorithms on the "open Gmail" task.

As shown in Figure 12, RL agents learn to solve the task via interactions in training environments resulting in their test performance increases. Still, we remark that RL agents fail to complete more complex tasks that require longer interactions for success. On the `Language` task, we find that these agents fail to complete the task even in training environments. While the agents enter the target application (i.e., the setting application) during exploration, they fail to further navigate the appropriate pages. This is mainly because the sparse success signal is only provided after completing the task, unable to guide the exploration from scratch.

### E.3 EFFECT OF REWARD SHAPING

Instead of directly using sparse success signals for reward, reward shaping can be a helpful direction to guide agents in the long-horizon task. To further study, we design a dense reward function in a `Call 911` task having an episode length of 6. The reward function is defined as a value of $i/N$ for accomplishing the $i$-th step of the total $N$ step task, and -1 otherwise. The value for $N$ and the completion criteria for each step are determined based on a human expert demonstration. We train the agent using DDQN with $\gamma = 0.1$ for 3500 episodes. We use a single training environment (with environment id 000).

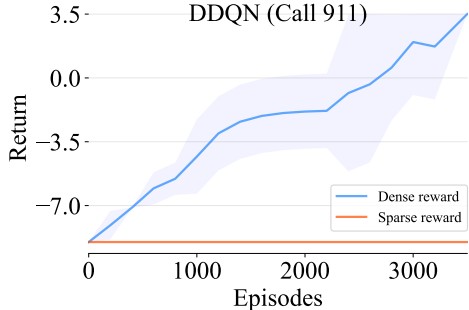

Figure 13: Training curves of DDQN algorithm on the "call 911" task.

Fig 13 displays the training curve with return values. The sum of the reward keeps increasing as the agent learns to progress each step. The return achieves the maximal value of $1/6 + \cdots + 6/6$, which indicates the success of the task in a training environment, while it fails in a slightly different test environment with id 100. We note that designing dense reward functions is an effortful process since it requires an understanding of UI elements per task step. The automation of the reward-shaping technique to many other tasks is highly expected to contribute to the success of reinforcement learning from scratch, suggesting a future direction toward more scalable reward function design techniques. We also expect that initializing the agent with a pre-trained policy (such as with a VLM encoder or LLM) can also improve the efficiency of RL.

