# OpenReview forum: "B-MoCA: Benchmarking Mobile Device Control Agents across Diverse Configurations"
_ICLR.cc/2025/Conference — Submitted to ICLR 2025_

### Official Review · Reviewer_QToU · 2024-11-01

**Soundness:** 3
**Presentation:** 3
**Contribution:** 1
**Rating:** 5
**Confidence:** 4

**Summary:**

This paper presents B-MoCA, a benchmark for evaluating mobile device control agents (LLMs) on diverse configurations. Built around the Android OS, with ADB support. B-MoCA encompasses 131 daily tasks with environment randomization features, such as varied icon placements, language settings, and device types. These tasks and configurations test agents' generalization abilities by benchmarking large language models (LLMs) and custom agents.

**Strengths:**

1. An open-source project is proposed to evaluate all the popular LLMs we have today.
2. B-MoCA addresses an important gap by focusing on environmental diversity, making it a useful resource for researchers aiming to improve agent robustness.

**Weaknesses:**

1. Since the work has already been proposed in an ICLR 2024 workshop, it lacks the fresh innovation expected of a full ICLR 2025 conference submission. This may restrict its perceived value in the community.
2. Lack of novelty, I know it is bad to just say this, but I truly can't get any insights from this paper. It's a well-built open-source project to evaluate LLMs indeed, but I did not find any conclusion here interesting.
3. Although several LLMs and MLLMs were benchmarked, the paper does not explore adaptive fine-tuning or other strategies to mitigate performance declines in randomized environments.

**Questions:**

Have any additional measures been tested to mitigate performance declines due to randomization?

**Details Of Ethics Concerns:**

This paper has been accepted by the ICLR 2024 workshop spotlight, and the updates after that are minor, most data and figures are identical.
https://arxiv.org/abs/2404.16660v1

---

> ### Author Response · Authors · 2024-11-25
>
> Dear reviewer QToU,
>
> Thank you for the time and effort in reviewing our work. We prepared responses to address your concerns one by one. Please feel free to leave further comments to address remaining concerns, if you have any.
>
> ---
>
> **[W1 - Updates from earlier version]**
>
> We highlight several updates compared to the earlier version. Above all, we have extended the number of tasks from 60 to 131 to incorporate a wider range of real-life usages, as the reviewer sGrp praises. Furthermore, we benchmark more diverse baselines, including agents built with open-source LLMs, as the reviewer g7ju remarks. We also provide extensive analyses regarding the additional experimental results, as the reviewer cuoT emphasizes.
>
> ---
>
> **[W2 - Novelty and impact of our work]**
>
> In this work, we have proposed a novel benchmark featuring environmental randomization to reflect diverse device configurations in real usages, which would be a key factor for real-world deployment. We emphasize that our contributions to the community include offering extensive experimental results and detailed analyses. These results include comparisons of the generalization abilities of proprietary LLMs versus state-of-the-art open-source LLMs, performance differences between text-only and multi-modal agents, the impact of using few-shot examples, and the role of environmental diversity in training agents for improved generalization. Furthermore, we emphasize that our findings reinforce strong consensus, aligning closely with what recent benchmarks have remarked. For example, these analyses include a limited multi-modal understanding of state-of-the-art LLM agents [1] and frequent hallucinations regarding task completion [2], offering valuable guidance for future research.
>
> [1] Christopher R., et al. (2024). AndroidWorld: A Dynamic Benchmarking Environment for Autonomous Agents.
>
> [2] Jingxuan C., et al. (2024). SPA-Bench: A Comprehensive Benchmark for SmartPhone Agent Evaluation.
>
> ---
>
> **[W3/Q1 - Methods for improving generalization ability]**
>
> We clarify that the main focus of our experiments is to provide a benchmark reference. As baselines, we have analyzed several methods for improving generalization abilities, including in-context learning with few-shot examples and fine-tuning parameters using the training dataset collected from diverse environments (please refer to Section 3.2 for a detailed explanation of these baselines). In our analyses, we also examine the effects of few-shot examples for proprietary LLMs and the diversity of environments for training in detail (please see Section 4.3 for more details).

---

> > ### Comment · Reviewer_QToU · 2024-11-26
> >
> > Thanks for your responses. I do value this work, but I still find no interesting insight for me. I will raise my score to 5.

---

### Official Review · Reviewer_cuoT · 2024-11-02

**Soundness:** 3
**Presentation:** 3
**Contribution:** 2
**Rating:** 5
**Confidence:** 3

**Summary:**

The paper introduces a novel benchmark for evaluating and developing mobile device control agents. Based on the Android operating system, B-MoCA defines 131 common daily tasks and incorporates a randomization feature to alter mobile device configurations to assess the generalization performance of agents. The evaluation is quite comprehensive.

**Strengths:**

Strengths
1. The proposed method is based on Android, ensuring authentic evaluation environments.
2. It includes 131 daily tasks grounded in realistic scenarios, covering a range of applications.
3. By changing various elements, it assesses the agents' generalization capabilities across various device configurations.
4. Open-source.
5. Comprehensive evaluation.

**Weaknesses:**

1. It is better to train a new model based on a training dataset collected in the same way. The collected dataset contains various environments, and it is natural to explore whether the agent could gain better generalization abilities when trained in such a dataset. However, the existing benchmark only evaluates existing models, and I think the contribution is not enough.
2. The conclusions are already well-known. The conclusion that I care about most is this one "such as their poor generalization in UI elements understanding and manipulation". This should be already well known at least in AppAgent, which serves as one of the code bases of this paper. In AppAgent, the authors utilize lots of tricks to improve the VLM's understanding abilities even using GPT-4V, such as generating documents and XML for references. Also, in GitHub, the author tested the best open-source VLM named Qwen-VL and reported a decrease in performance. All these could reflect the fact that VLM's limitations in understanding, and also prove that open-source VLMs are weaker than close-source ones.
3. Could you elaborate on the reasons for evaluating open-source VLMs alongside closed-source models, considering that there is a well-known performance gap between them?

**Questions:**

see the weaknesses.

---

> ### Author Response · Authors · 2024-11-25
>
> Dear reviewer cuoT,
>
> We appreciate your efforts and time in reviewing our work. Regarding several questions and concerns you have raised, we prepared responses one by one. Please feel free to discuss further to address remaining concerns, if you have any.
>
> ---
>
> **[W1 - Clarification for baseline with training]**
>
> We clarify that the **custom** agents we explore are trained by using datasets collected in training environments, as you suggested. Our analysis, which examines the effect of the number of environments used for dataset collection on the generalization ability of these agents, is available in Section 4.3.
>
> ---
>
> **[W2 - Clarification on contributions in our conclusions]**
>
> We emphasize that our analyses regard generalization ability over environmental randomization as device configuration changes. These results include comparisons of the **generalization abilities across various device settings** (e.g., diverse language and device types, on top of unseen apps that AppAgent explored) of proprietary LLMs versus state-of-the-art open-source LLMs, performance differences between text-only and multi-modal agents, the impact of using few-shot examples, and the role of environmental diversity in training agents for improved generalization.
>
> ---
>
> **[W3 - Rationale behind evaluating open-source models]**
>
> The main rationale stems from the several challenges of leveraging closed-source models, such as in fine-tuning, as stated in Section 3.2 (line 245). We find that incorporating training while developing agents can be highly effective in our six selective challenging tasks. As discussed in Section 4.2 (starting from lines 422-423), we observe that these agents show success rates surpassing the agents exploiting proprietary LLMs on several tasks. We believe that this clearly shows that training agents can be a promising way. Furthermore, we analyze the effect of the number of environments used for dataset collection in Section 4.3, providing insights with regard to the generalization ability when developing agents based on open-source models via training.

---

> > ### Comment · Area_Chair_jtL7 · 2024-11-30
> >
> > Dear Reviewer,
> >
> > The authors have provided their responses. Could you please review them and share your feedback?
> >
> > Thank you!

---

> > ### Comment · Reviewer_cuoT · 2024-12-02
> > **Official comment by reviewer cuoT**
> >
> > Thanks for your clarification. I will keep my score.

---

### Official Review · Reviewer_g7ju · 2024-11-03

**Soundness:** 2
**Presentation:** 3
**Contribution:** 1
**Rating:** 5
**Confidence:** 4

**Summary:**

This paper introduces the B-MoCA benchmark with interactive environments for evaluating and developing mobile device control agents. Then several baseline agents for mobile device control are evaluated, identifying their limitations, such as their poor generalization in UI elements understanding and manipulation.

**Strengths:**

1. B-MoCA introduces a well-structured benchmark for evaluating mobile control agents.
2. B-MoCA contains 131 common daily tasks, from simple tasks to complex multi-step interactions to assess essential skills for mobile device management.
3. The benchmark tests agents based on various models and environment randomization.

**Weaknesses:**

1.  This paper lacks contribution and novelty, primarily presenting a benchmark for evaluating mobile device control agents. The conclusions drawn from the experiments lack depth, making it challenging to derive valuable insights for further improvement.
2. The paper lacks a qualitative comparison with other benchmarks for decision-making agents. Demonstrating that the proposed benchmark is more effective or of higher quality than existing alternatives is crucial to establishing its value.

**Questions:**

no

**Details Of Ethics Concerns:**

The paper was published in the ICLR 2024 Workshop on Generative Models for Decision Making, and the submitted version has only minor differences from the workshop paper. I'm uncertain whether it still qualifies for publication in this conference.

---

> ### Author Response · Authors · 2024-11-25
>
> Dear reviewer g7ju,
>
> We highly value your time and efforts in reviewing our manuscript. We have prepared responses to each concern you have raised. These responses include clarification of the contributions in our findings and the contribution of our work compared to existing benchmarks. Please feel free to leave more comments for further discussion.
>
> ---
> **[W1 - Contribution of B-MoCA]**
>
> In this work, we have proposed a novel benchmark featuring environmental randomization to reflect diverse device configurations in real usages, which would be a key factor for real-world deployment. We emphasize that our contributions to the community include offering extensive experimental results and detailed analyses. These results include comparisons of the generalization abilities of proprietary LLMs versus state-of-the-art open-source LLMs, performance differences between text-only and multi-modal agents, the impact of using few-shot examples, and the role of environmental diversity in training agents for improved generalization. Furthermore, we emphasize that our findings reinforce strong consensus, aligning closely with what recent benchmarks have remarked. For example, these analyses include a limited multi-modal understanding of state-of-the-art LLM agents [1] and frequent hallucinations regarding task completion [2], offering valuable guidance for future research.
>
> ---
>
> **[W2 - Comparison with existing benchmarks]**
>
> Thanks for raising an important point. Following your suggestion, we provide a detailed comparison of our benchmark platform with other existing works in the table below. We plan to include this comparison in the revised version of the manuscript.
>
> | Feature                           | AndroidEnv [3] | MobileEnv [4] | AndroidWorld [1] | SPA-Bench [2]  | B-MoCA (ours) |
> |------------------------------|:-------------------:|:------------------:|:---------------------:|:-------------------:|:-------------------:|
> | Realistic tasks                | x                        | o                        | o                           | o                        | o                         |
> | Types of agents evaluated    | Custom       | LLM/MLLM     | LLM/MLLM       | LLM/MLLM/Custom  | LLM/MLLM/Custom  |
> | Analysis with various device settings | x   | x                       | x                           | x                         | o                        |
> | Evaluation pipeline with varying devices  | x        | x           | x                           | x                         | o                        |
> | Open-sourced training pipeline    | o         | x                         | x                           | x                         | o                       |
>
> [1] Christopher R., et al. (2024). AndroidWorld: A Dynamic Benchmarking Environment for Autonomous Agents.
>
> [2] Jingxuan C., et al. (2024). SPA-Bench: A Comprehensive Benchmark for SmartPhone Agent Evaluation.
>
> [3] Daniel T., et al.,(2021). AndroidEnv: A Reinforcement Learning Platform for Android.
>
> [4] Danyang Z., et al. (2023). Mobile-Env: Building Qualified Evaluation Benchmarks for LLM-GUI Interaction.

---

> > ### Comment · Reviewer_g7ju · 2024-11-25
> >
> > Thank you for the author's responses. I would like to keep my score as the contribution of this work seems limited. Additionally, there is a lack of quantitive comparisons between your benchmark platform and existing methods.

---

### Official Review · Reviewer_sGrp · 2024-11-04

**Soundness:** 3
**Presentation:** 3
**Contribution:** 2
**Rating:** 5
**Confidence:** 5

**Summary:**

The paper presents a commendable and promising benchmark, B-MoCA, designed to evaluate mobile device control agents across a variety of device configurations. The task design is somewhat aligned with real-world scenarios, encompassing a broad spectrum of everyday activities. Additionally, the incorporation of environment randomization enhances the diversity of testing conditions. Overall, the presentation of the paper is clear, and the figures effectively support the data and findings.

**Strengths:**

B-MoCA defines 131 common daily tasks, such as opening applications, conducting web searches, and adjusting device settings. These tasks are grounded in realistic scenarios, making the benchmark more reflective of actual user needs than previous benchmarks that focused on simpler tasks. The authors emphasize the environment randomization feature, which introduces variations in device configurations (e.g., icon locations, languages, device types). This setup broadens the testing scope for agents, helping to prevent scenarios where agents merely "memorize" operations in fixed environments. Such variability facilitates the assessment of agent adaptability across different configurations, contributing to a more robust evaluation of agent generalization.

**Weaknesses:**

1. **Comparison with Existing Benchmarks:**

   While B-MoCA increases the task count and introduces environment randomization, it does not fundamentally differ in its core framework or testing methodology from existing benchmarks like WebShop and Mobile-Env. For instance, Mobile-Env provides a platform for training and evaluating mobile agents with a focus on GUI interaction, supporting both visual-based and text-based agents. [GitHub](https://github.com/X-LANCE/Mobile-Env) Similarly, WebShop offers a scalable environment for web-based interactions, emphasizing language grounding and decision-making. [Webshop Pnlp](https://webshop-pnlp.github.io/) B-MoCA appears to serve as an extension of these benchmarks rather than introducing a novel evaluation paradigm.  Could the authors provide a more detailed comparison between B-MoCA and existing benchmarks like WebShop and Mobile-Env, highlighting specific novel aspects or improvements in B-MoCA's methodology or evaluation framework?

2. **Multimodal Model Challenges:**

   Although B-MoCA accommodates multimodal models (such as those with image-based inputs), the paper does not thoroughly explore the specific challenges and limitations of these models in visual comprehension. For instance, the inclusion of multimodal inputs heightens model computational complexity, potentially resulting in latency issues, which are particularly critical in mobile device contexts. Recent studies have highlighted the importance of optimizing multimodal models for mobile devices to address these challenges. [ArXiv](https://arxiv.org/abs/2405.12107)    Could the authors discuss how B-MoCA addresses the computational complexity and potential latency issues associated with multimodal models, particularly in the context of mobile devices? Are there specific metrics or evaluations in B-MoCA designed to assess these aspects?

3. **Evaluation Scope:**

   The evaluation focuses solely on the capabilities of large language models (LLMs), while many open-source GUI agents, such as [1,2,3,4], remain unassessed by this benchmark. This limitation complicates the evaluation of future works utilizing B-MoCA. Including these could provide a more comprehensive comparison and facilitate easier adoption of B-MoCA for future research.

[1] AutoDroid，
[2] You Only Look at Screens，
[3] CoCo-Agent
[4] AppAgent

**Questions:**

Please refer to the weaknesses section above for specific points requiring clarification or further detail.

---

> ### Author Response · Authors · 2024-11-25
>
> Dear reviewer sGrp,
>
> We are thankful for your time and efforts in reviewing our manuscript. Regarding several concerns you have raised, we have prepared responses, including additional experiments. We also have clarified the strength of our work compared to existing benchmarks. Please feel free to discuss further to address the remaining concerns, if you have any.
>
> ---
>
> **[W1 - Comparison with existing benchmarks]**
>
> Thanks for raising an important question. While our benchmark is grounded on a commonly used framework defining mobile device control as a sequential decision-making problem, our work serves unique and important points. Because our main focus is on evaluating and developing agents with varying device configurations, our framework features an easily executable platform for configuring diverse environments with differing devices (please refer to supplementary code materials), the training and evaluation pipeline regarding those changes, and detailed analyses of the generalization ability of baseline agents on these varying environments. We provide a detailed comparison of our benchmark platform with other existing works, including Web-Shop [1] and MobileEnv [3].
>
> | Feature                                 | Web-Shop [1]   | AndroidEnv [2] | MobileEnv [3] | AndroidWorld [4] | SPA-Bench [5]  | B-MoCA (ours) |
> |-----------------------------------|:------------------:|:-------------------:|:------------------:|:--------------------:|:--------------------:|:-------------------:|
> | Target platform                   | Desktop Web     | Android (apps)   | Android (apps) | Android (apps)   | Android (apps)   | Android (apps) |
> | Realistic tasks                      | o                       | x                         |  o                       | o                          | o                          | o                       |
> | Types of agents evaluated    | Custom            | Custom               | LLM/MLLM    | LLM/MLLM      | LLM/MLLM/Custom  | LLM/MLLM/Custom |
> | Analysis with various device settings | x       | x                         | x                         | x                         | x                          | o                       |
> | Evaluation pipeline with varying devices | x      | x               | x                           | x                            | x                           | o                    |
> | Open-sourced training pipeline    | o              | o                      | x                         | x                            | x                           | o                    |
>
>
> [1] Shunyu Y, et al. (2022). WebShop: Towards Scalable Real-World Web Interaction with Grounded Language Agents
>
> [2] Daniel T., et al.,(2021). AndroidEnv: A Reinforcement Learning Platform for Android.
>
> [3] Danyang Z., et al. (2023). Mobile-Env: Building Qualified Evaluation Benchmarks for LLM-GUI Interaction.
>
> [4] Christopher R., et al. (2024). AndroidWorld: A Dynamic Benchmarking Environment for Autonomous Agents.
>
> [5] Jingxuan C., et al. (2024). SPA-Bench: A Comprehensive Benchmark for SmartPhone Agent Evaluation.
>
>
> ---
>
> **[W2 - More analysis on MLLM agents]**
>
> Following your nice suggestion, we additionally conducted an experiment analyzing the limitation of MLLM agents, compared to LLM agents, regarding the time complexity. We measure the average time consumed for the agents to make each decision across all steps in the episodes, solving six challenging representative tasks using one test environment. We provide exact values in the table below. We find that the MLLM agents take longer time compared to LLM agents on average. We note that considering additional methods for addressing the limitation in computational complexity is beyond our main focus, while we believe that our findings provide valuable insights and strong consensus to the community when developing new algorithms.
>
> | Model               | Average Time (seconds) |
> |---------------------|-------------------------|
> | GPT-4o (multi-modal)   | 2.03        |
> | GPT-4o (text-only)     | 1.78           |
> | Gemini-1.5-pro (multi-modal)| 2.92      |
> | Gemini-1.5-pro (text-only)  | 2.12         |

---

> > ### Author Response · Authors · 2024-11-25
> >
> > **[W3 - On baselines]**
> >
> > Thanks for pointing out this point. We clarify that the baselines we experiment are based on or already share many components discussed in the suggested works [6,7,8,9]. For example, the custom agents using the VLM encoder are the fine-tuned version of Auto-GUI [7]. Also, similar to exploration demonstrations collected for in-context learning of AutoDroid [6] and AppAgent [9], the few-shot examples used for MLLM agents in our experiment provide useful guidelines but with higher relevance to the tasks. We plan to include these clarifications with a more detailed comparison in our manuscript to reduce the complications for future work.
> >
> > To serve as a better reference, we additionally experiment with M3A [4] in our six selected challenging representative tasks. M3A is currently the state-of-the-art GUI agent, featuring a high understanding of the context while performing tasks by prompting the agent to summarize each step of action and the corresponding effect. We experiment with M3A (a11y tree), following the best-performing version on Android tasks in the original paper. We find that M3A achieves the highest performances among MLLM agents using GPT-4o (either zero-shot or few-shot). The auxiliary history of context helps agents in many tasks even better than the few-shot examples of human expert demonstrations (e.g., Calculator and Wikipedia). However, we observe that the agents using fine-tuned open-source LLMs with the human expert demonstrations (Custom agent using fine-tuned Llama-3-8B) still outperform in some tasks (e.g., Alarm (complex) and Call). Please refer to the table below for the exact success rate values for each task. We plan to include these in our next version of the manuscript.
> >
> > |                 | M3A (GPT-4o) | MLLM agent  (GPT-4o; zero-shot) | MLLM agent (GPT-4o; few-shot) | Custom agent (Llama-3-8B) |
> > |-----------------|:------------:|:-------------------------------:|:-----------------------------:|:-------------------------:|
> > | Alarm (simple)  |    70 ± 05   |             80 ± 06             |            63 ± 09            |          80 ± 00          |
> > | Alarm (complex) |    27 ± 03   |             23 ± 03             |            47 ± 12            |          80 ± 00          |
> > | Language        |    67 ± 07   |             53 ± 03             |            77 ± 03            |          10 ± 00          |
> > | Call            |    17 ± 05   |             00 ± 00             |            23 ± 13            |          83 ± 03          |
> > | Calculator      |    47 ± 07   |             13 ± 03             |            07 ± 07            |          73 ± 03          |
> > | Wikipedia       |    43 ± 05   |             27 ± 03             |            20 ± 06            |          10 ± 00          |
> > | Average         |    45 ± 08   |             33 ± 01             |            39 ± 03            |          56 ± 01          |
> >
> > [6 - AutoDroid] Hao W., et al. (2023). AutoDroid: LLM-powered Task Automation in Android.
> >
> > [7 - You Only Look at Screens] Zhousheng Z., et al. (2023). You Only Look at Screens: Multimodal Chain-of-Action Agents.
> >
> > [8 - CoCo-Agent] Xinbei M., et al. (2024). CoCo-Agent: A Comprehensive Cognitive MLLM Agent for Smartphone GUI Automation.
> >
> > [9 - AppAgent] Chi Z., et al. (2023). AppAgent: Multimodal Agents as Smartphone Users.

---

> > > ### Comment · Area_Chair_jtL7 · 2024-11-30
> > >
> > > Dear Reviewer,
> > >
> > > The authors have provided their responses. Could you please review them and share your feedback?
> > >
> > > Thank you!

---

> > > ### Comment · Reviewer_sGrp · 2024-11-30
> > >
> > > Thank you for the author's response. I would like to see the results of existing agents tested on your proposed benchmark, rather than just testing some related components. Without these tests, future research will not be able to demonstrate the advantages over past works by reporting results on your benchmark. In summary, I believe it is necessary to test some existing agents on your benchmark directly. Therefore, I will maintain my score.

---

### Official Review · Reviewer_zFWV · 2024-11-08

**Soundness:** 3
**Presentation:** 2
**Contribution:** 2
**Rating:** 5
**Confidence:** 5

**Summary:**

This paper introduces B-MoCA, a benchmark designed to evaluate mobile device control agents. It includes 131 common daily tasks and incorporates a randomization feature to test agents’ generalization across diverse device configurations, such as icon placements, wallpapers, languages, and device types. The benchmark evaluates various agents, including closed/open-source (M)LLMs and custom models trained from scratch. The paper shows agents' proficiency in straightforward tasks, but they struggle with complex tasks requiring multiple interactions. The paper provides a unified platform for comparing different methods, identifies limitations in current approaches, and offers open-source resources for reproducibility.

**Strengths:**

The benchmark includes a wide range of device configurations, providing a novel and important testing environment for the generalization capabilities of mobile device agents. Grounded in real-world applications, the benchmark is highly relevant for practical use. Its scope includes agents from a diverse set of models.

**Weaknesses:**

The range of tasks and mobile apps tested is somewhat narrow, which may not fully represent the potential capabilities of mobile device control agents. The paper lacks a compelling conclusion and fails to present inspiring findings that could significantly advance the field of research. This paper does not study the latest research and SoTA methods in mobile device agents, particularly those featuring more complex designs such as multi-agent systems and advanced UI understanding modules.

**Questions:**

The benchmark could be expanded to include more diverse and complex tasks that reflect real-world scenarios more accurately. Incorporating a study of SoTA methods and agents would provide more impactful findings. This would not only enhance the benchmark’s comprehensiveness but also offer deeper insights into the capabilities and limitations of current mobile device control agents.

---

> ### Author Response · Authors · 2024-11-25
>
> Dear reviewer zFWV,
>
> We highly appreciate your time and efforts in reviewing our work. We have considered your valuable comments and prepared responses to address your concerns one by one. We have also followed several suggestions and conducted additional experiments.
>
> ---
>
> **[W1/Q1 - Range of tasks]**
>
> Thanks for pointing out an important aspect of our work. While creating tasks, we prioritized building simple but regular tasks, without losing their connection to daily life, where the changes in the UIs associated with the tasks are meaningful to challenge the agents. This is because the main focus of our benchmark is to evaluate and develop agents that can understand and manipulate UI elements in mobile devices **on varying device configurations**. We believe our experimental results demonstrate that there is considerable room for the state-of-the-art models to handle the existing challenges, indicating the complexity of our current set of tasks.
>
> To reflect real-world scenarios more accurately, we plan to extend the set of tasks to include a broader set of tasks with more applications in the future version. We highlight that we also carefully design our open-source platform so that other users can easily incorporate new tasks.
>
> ---
>
> **[W2 - Contribution of findings in B-MoCA]**
>
> We emphasize that our contributions to the community encompass extensive experimental results and detailed analyses regarding environmental randomization, which is a key factor in real-world deployment. These results include comparisons of the generalization abilities of proprietary LLMs versus state-of-the-art open-source LLMs, performance differences between text-only and multi-modal agents, the impact of using few-shot examples, and the role of environmental diversity in training agents for improved generalization. Furthermore, we emphasize that our findings reinforce strong consensus, aligning closely with what recent benchmarks have remarked. For example, these analyses include a limited multi-modal understanding of state-of-the-art LLM agents [1] and frequent hallucinations regarding task completion [2], offering valuable guidance for future research.
>
> [1] Christopher R., et al. (2024). AndroidWorld: A Dynamic Benchmarking Environment for Autonomous Agents.
>
> [2] Jingxuan C., et al. (2024). SPA-Bench: A Comprehensive Benchmark for SmartPhone Agent Evaluation.
>
> ---
>
> **[W3/Q2 - Benchmark reference with state-of-the-art methods]**
>
> Thanks for the nice suggestion. To improve our benchmark reference, we have conducted additional experiments with the M3A agent [1]. M3A is currently the state-of-the-art GUI agent, featuring a high understanding of the context while performing tasks by prompting the agent to summarize each step of action and the corresponding effect. We experiment with M3A (a11y tree), following the best-performing version on Android tasks in the original paper. We find that M3A achieves the highest performances among MLLM agents using GPT-4o (either zero-shot or few-shot). The auxiliary history of context helps agents in many tasks even better than the few-shot examples of human expert demonstrations (e.g., Calculator and Wikipedia). However, we observe that the agents using fine-tuned open-source LLMs with the human expert demonstrations (Custom agent using fine-tuned Llama-3-8B) still outperform in some tasks (e.g., Alarm (complex) and Call). Please refer to the table below for the exact success rate values for each task. We plan to include these in our next version of the manuscript.
>
> |                 | M3A (GPT-4o) | MLLM agent  (GPT-4o; zero-shot) | MLLM agent (GPT-4o; few-shot) | Custom agent (Llama-3-8B) |
> |-----------------|:------------:|:-------------------------------:|:-----------------------------:|:-------------------------:|
> | Alarm (simple)  |    70 ± 05   |             80 ± 06             |            63 ± 09            |          80 ± 00          |
> | Alarm (complex) |    27 ± 03   |             23 ± 03             |            47 ± 12            |          80 ± 00          |
> | Language        |    67 ± 07   |             53 ± 03             |            77 ± 03            |          10 ± 00          |
> | Call            |    17 ± 05   |             00 ± 00             |            23 ± 13            |          83 ± 03          |
> | Calculator      |    47 ± 07   |             13 ± 03             |            07 ± 07            |          73 ± 03          |
> | Wikipedia       |    43 ± 05   |             27 ± 03             |            20 ± 06            |          10 ± 00          |
> | Average         |    45 ± 08   |             33 ± 01             |            39 ± 03            |          56 ± 01          |

---

> > ### Comment · Reviewer_zFWV · 2024-11-26
> >
> > Thanks for your responses. I will increase the score from 3 to 5. However I think the comparisons with related benchmarks are still limited. And the insights from this benchmarks paper also need some improvements.

---

### Meta-Review · Area_Chair_jtL7 · 2024-12-14

**Metareview:**

The paper introduces B-MoCA, a large-scale benchmark for evaluating mobile device control agents across diverse device configurations in a wide range of everyday activities.

The paper is well-written, and the scale of the experiments is commendable. Additionally, the commitment to releasing code and models enhances reproducibility.

However, all reviewers consistently rated the paper below the acceptance threshold, with a score of 5. Three key concerns were identified:

Lack of comparison with existing benchmarks, such as WebShop and Mobile-Env.

Insufficient comparison with existing agents, including several open-source GUI agents (e.g., [1,2,3,4]), as highlighted by Reviewer sGrp.

Limited insights from experiments, as the conclusion that models struggle with generalizing UI element understanding and manipulation is already well-established in prior work like AppAgent.

As these issues remain unaddressed, the AC has decided to reject the paper. The authors are encouraged to address these concerns in a revised version and consider resubmitting to future conference venues.

**Additional Comments On Reviewer Discussion:**

All reviewers consistently rated the paper below the acceptance threshold, with a score of 5. Three key concerns were identified:

Lack of comparison with existing benchmarks, such as WebShop and Mobile-Env.

Insufficient comparison with existing agents, including several open-source GUI agents (e.g., [1,2,3,4]), as highlighted by Reviewer sGrp.

Limited insights from experiments, as the conclusion that models struggle with generalizing UI element understanding and manipulation is already well-established in prior work like AppAgent.

As these issues remain unaddressed even after the rebuttal, the AC has decided to reject the paper. The authors are encouraged to address these concerns in a revised version and consider resubmitting to future conference venues.

---

### Decision · Program_Chairs · 2025-01-22

Reject